# BTBS-LNS: Binarized-Tightening, Branch and Search on Learning LNS Policies for MIP

**Hao Yuan[1], Wenli Ouyang[1]***, **Changwen Zhang[1], Yong Sun[1], Liming Gong[1], Junchi Yan[2]**

[1]AI Lab, Lenovo Research

[2]School of Artificial Intelligence & Department of Computer Science and Engineering & MoE Lab of AI, Shanghai Jiao Tong University

`{yuanhao4, ouyangwl1, zhangcw5, sunyong4, gonglm3}@lenovo.com,`
`yanjunchi@sjtu.edu.cn`

## Abstract

Learning to solve large-scale Mixed Integer Program (MIP) problems is an emerging research topic, and policy learning-based Large Neighborhood Search (LNS) has been a popular paradigm. However, the explored space of LNS policy is often limited even in the training phase, making the learned policy sometimes wrongly fix some potentially important variables early in the search, leading to local optimum in some cases. Moreover, many methods only assume binary variables to deal with. We present a practical approach, termed Binarized-Tightening Branch-and-Search for Large Neighborhood Search (BTBS-LNS). It comprises three key techniques: 1) the "Binarized Tightening" technique for integer variables to handle their wide range by binary encoding and bound tightening; 2) an attention-based tripartite graph to capture global correlations among variables and constraints for an MIP instance; 3) an extra branching network as a global view, to identify and optimize wrongly-fixed backdoor variables at each search step. Experiments show its superior performance over the open-source solver SCIP and LNS baselines. Moreover, it performs competitively with, and sometimes better than the commercial solver Gurobi (v9.5.0), especially on the MIPLIB2017 benchmark chosen by Hans Mittelmann, where our method can deliver 10% better primal gaps compared with Gurobi in a 300s cut-off time.

## 1 Introduction and Related Work

Mixed-integer programming (MIP) is a well-established optimization problem. In many cases, feasible or even optimal solutions are required under strong time limits, and thus efficiently finding high-quality solutions is of great importance. Recently, machine learning for combinatorial optimization has been an emerging topic (Bengio et al., 2021) with prominent success in different tasks, e.g. graph matching (Yan et al., 2020), and ML4MIP is also an emerging field (Zhang et al., 2023).

A variety of deep learning-based solving methods were proposed to deal with specific MIP problems, including construction methods (Ma et al., 2019; Xing & Tu, 2020; Fu et al., 2021; Zhang et al., 2020; Khalil et al., 2017; Xin et al., 2021) and iterative-based refinements (Wu et al., 2021b; Chen & Tian, 2019; Lu et al., 2019; Li et al., 2020). While they cannot be directly applied to a wider scope of MIP problems, and thus learning the solving policies for general MIP problems has also been intensively studied, in which the primal heuristics catch more attention, including Large Neighborhood Search (LNS) (Wu et al., 2021a; Song et al., 2020; Nair et al., 2020a) and Local Branching (LB) (Liu et al., 2022). This paper focuses on LNS for solving general MIP problems – the powerful yet expensive iteration-based heuristics (Hendel, 2022).

Traditional LNS methods usually explore a complex neighborhood by predefined heuristics (Gendreau et al., 2010), in which the heuristic selection is a long-standing challenging task, especially for general MIP problems, which may require heavy efforts to design valid heuristics. Learning-based methods provide a possible direction. Both Imitation Learning (IL) (Song et al., 2020) and

---

*Corresponding author.

Table 1: Comparison of our method to existing works, and it achieves the SOTA performance.

| References | Applicability | Approach | Addressing Local Optima | Training |
|---|---|---|---|---|
| Huang et al. (2023b) | Binary | LNS | Adaptive Neighborhood Size | Contrastive Learning |
| Liu et al. (2022) | Binary | Local Branching | RL-based Branching Size | Regression + RL |
| Wu et al. (2021a), Nair et al. (2020a) | Binary | LNS | / | RL |
| Song et al. (2020) | Binary | LNS | / | Imitation & RL |
| Hendel (2022) | General MIP | ALNS (Heuristic in B&B) | Adaptive Control for Multiple Heuristics | Multi-armed Bandit |
| Sonnerat et al. (2021) | General MIP | LNS | Adaptive Neighborhood Size | Imitation |
| **BTBS-LNS (Ours)** | General MIP | Branching on top of LNS | **Step-wise Global Information** | **RL (LNS) + Imitation (Branching)** |

Reinforcement Learning (RL) (Wu et al., 2021a; Nair et al., 2020a) showed effectiveness in learning decomposition-based LNS policies. However, there are still some challenges. The performance of the learned policies may significantly degrade when applied to general integers due to the vast scale of candidate values (compared to binary variables), leading to a large complexity in optimization. Moreover, the learned policies may be trapped in local optimum for complicated cases.

In this paper, we propose a Binarized-Tightening, Branch and Search-based LNS approach (**BTBS-LNS**) for general MIP problems. Specifically, we design the "Binarized Tightening" algorithm to deal with the optimization for general integer variables, where we first binarize the general integer variables and express them with the resulting bit sequence, and then tighten the bound of original variables w.r.t. the LNS decision along with the current solution. In this way, the variable bounds can be tightened and explored effectively at a controlled complexity. Based on our binarization formulation, we further employed an attention-based tripartite graph (Ding et al., 2020) to encode the MIP instances and improved the attention architecture by removing the softmax normalization, which allows us to fully preserve the raw weights between neighboring nodes. Meanwhile, to enhance exploration and optimize some wrongly-fixed backdoor variables (Williams et al., 2003; Khalil et al., 2022) by the learned LNS policy, we leverage an extra branching network at each step, providing branching decisions at the global view [1] to help escape local optimum. In a nutshell, this paper can be characterized by the following bullets, which we believe are common building blocks:

**1) Bound Tightening for MIP.** We propose a new "Binarized Tightening" scheme for general MIP problems with an efficient embodiment of variable encoding and bound tightening techniques.

**2) Combining global information step by step with LNS.** To assist the learned policy in escaping the local optimum efficiently, we devise an extra variable branching mechanism to select and optimize the LNS wrongly fixed backdoor variables, by contrast with the global optimum. The hybrid branch and search policy greatly enhance exploration and show efficiency.

**3) Problem encoding with improved attention architecture.** We employ an attention-based tripartite graph to encode MIP problems and capture correlations using an improved attention approach, which demonstrates empirical effectiveness.

**4) Strong empirical results.** Experiments on seven MIP problems show that our method consistently outperforms the LNS baselines and open-source SCIP (Gamrath et al., 2020). On MIPLIB2017 benchmark[2] chosen by Hans Mittelmann, it even achieves superior performance over Gurobi, purely taking SCIP as the baseline solver. It can further boost Gurobi when taking Gurobi as the baseline solver (see Appendix A.4).

We summarize the key related works in Table 1, and elaborate on more details in Appendix A.1.

## 2 PRELIMINARIES

**Mixed Integer Program (MIP)** is in general defined as:

$$
\begin{aligned}
\min \quad & \mathbf{c}^\top \mathbf{x} \\
s.t. \quad & \mathbf{A}\mathbf{x} \le \mathbf{b} \\
& x_i \in \{0,1\}, \forall i \in \mathcal{B}; x_j \in Z^+, \forall j \in \mathcal{G}; x_k \ge 0, \forall k \in \mathcal{C}
\end{aligned}
\tag{1}
$$

where $\mathbf{x} \in \mathbb{R}^n$ is a vector of $n$ decision variables; $\mathbf{c} \in \mathbb{R}^n$ denotes the vector of objective coefficients. $\mathbf{A}\mathbf{x} \le \mathbf{b}$ denotes the overall $m$ linear constraints, where $\mathbf{A} \in \mathbb{R}^{m \times n}$ represents the incidence

---

[1]A broader context that goes beyond the immediate LNS observations, e.g., contrast with global optimum.
[2]https://plato.asu.edu/bench.html

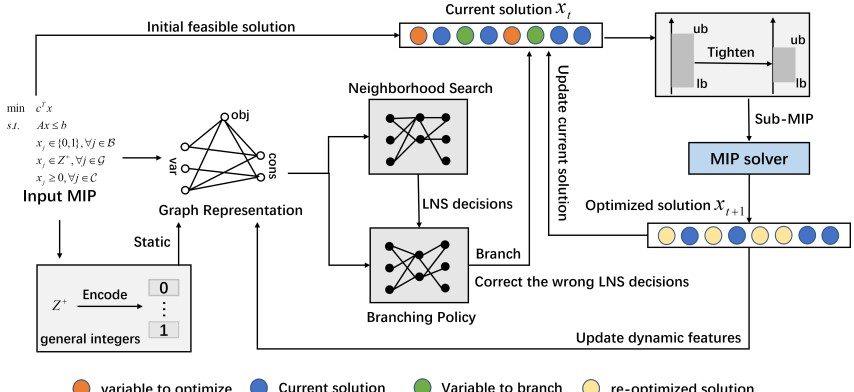

Figure 1: Overview of **BTBS-LNS**. First, we propose "Binarize Tightening" to handle general integer variables. The *Binarize* mechanism can binary-encode the variables and split them into sub-optimization bits. With the bit-wise decision by LNS, the variable bounds can be refined by bound tightening. Second, we devise a branching network on top of LNS to select wrongly fixed backdoor variables at a global view, which may help efficiently escape local optimum in some cases.

matrix, with $\mathbf{b} \in \mathbb{R}^m$. For general MIP instances, the index set of $n$ variables $\mathcal{N} := \{1, ..., n\}$ can be partitioned into three sets, binary variable set $\mathcal{B}$, general integer variable set $\mathcal{G}$ and continuous variable set $\mathcal{C}$. MIP presents greater challenges compared to integer programming (Wu et al., 2021a) as the continuous variables may require distinct optimization policies with integer variables.

**Large Neighborhood Search (LNS)** is a powerful yet expensive heuristic (Gendreau et al., 2010). It takes the best solution so far $\mathbf{x}^*$ as input and searches for the local optimum in its neighborhood:

$$\mathbf{x}' = \arg\min_{\mathbf{x} \in N(\mathbf{x}^*)} \{\mathbf{c}^\top \mathbf{x}\} \tag{2}$$

where $N(\cdot)$ is a predefined neighborhood - the search scope at each step, and $\mathbf{x}'$ denotes the optimized solution within $N(\mathbf{x}^*)$, obtained by destroying and re-optimization from the current solution.

Compared to local search heuristics, LNS can be more effective by using a broader neighborhood. However, the selection of neighborhood function $N(\cdot)$ is nontrivial. Heuristic methods mainly rely on problem-specific operators, e.g., 2-opt (Flood, 1956) in TSP, which call for considerable trial-and-error and domain knowledge (Papadimitriou & Steiglitz, 1998). The currently popular learning-based approaches mainly focus on binary variables and may be trapped in local optimum due to the learning complexity. In this paper, we propose a binarized-tightening branch-and-search LNS approach, designed to address general MIP problems. It may efficiently escape local optimum when the LNS decisions are unreliable in some scenarios.

## 3 METHODOLOGY

### 3.1 OVERVIEW

Fig. 1 presents the overview of our approach. The input is a MIP instance, with its initial feasible solution $\mathbf{x}_0$ generated by a baseline solver. General integer variables are first encoded into binary substitute variables, and the instance is subsequently represented as a tripartite graph (Ding et al., 2020), which is then fed into the large neighborhood search network, selecting the variable subsets that may need to be optimized at each step, with the remaining variables fixed or bound-tightened (**see Sec. 3.2 and 3.3**). Additionally, we devise an extra branching network to select some wrongly-fixed backdoor variables by the learned LNS policy, to help escape local optimum. With the sequential decisions of the branch and search policy and the resulting tightened variable bounds, an off-the-shelf solver, e.g. SCIP, is applied to obtain the optimized feasible solution $\mathbf{x}_{t+1}$. Iterations continue until the time limit is reached, and the optimized solutions can be obtained.

In general, the neighborhood search policy and branching policy are trained sequentially, where the training details are described in Sec. 3.3 and 3.4, respectively. They optimize the current solution from different views and may remedy the local search drawbacks in some cases.

## 3.2 THE BINARIZED TIGHTENING SCHEME

Variables in general MIP instances can be divided into three categories: binary, general integer (with arbitrary large value), and continuous variables. Previous studies mainly focused on the binary variables ($0/1$). Limited values greatly simplify the optimization, making it easier to deal with compared to the general integer variables, and some learning frameworks have proved their effectiveness (Wu et al., 2021a; Song et al., 2020). In this paper, we concentrated on more general MIP problems, especially for general integer variables.

An intuitive method is to directly migrate some efficient binary LNS approaches, e.g., Wu et al. (2021a), to general integers. In this way, different types of variables are equally treated. At each step, we fix some of the variables (no matter what type the variable belongs to), and solve the sub-MIP with a baseline solver e.g. SCIP (Gamrath et al., 2020) or Gurobi (G., 2020). However, empirical results revealed that the simplified generalized LNS approach (e.g., RL-LNS (Wu et al., 2021a)) is much slower and significantly underperforms the MIP solvers, e.g., Gurobi. (see Table 4 and Fig. 3 for detail comparison.)

To address these challenges, we propose the so-called "Binarized Tightening" scheme for MIP. The idea is to confine the variables within a narrow range around the current solution, rather than directly fixing them, to balance exploration and exploitation. It shares similar insights with local search, which relies on the current best solution to guide the search, thus avoiding blind search throughout the entire solution space. Specifically, we represent each general integer variable with $d = \lceil \log_2 (ub - lb) \rceil$ binary variables at a decreasing magnitude, where $ub$ and $lb$ are the upper and lower bounds of the original variable, respectively. The subsequent optimization is applied to the substitute binary variables, indicating whether the current solution is reliable or not. In this way, we transform the LNS for the original variable into multiple decisions on substitution variables. Note that the unbounded variables where $ub$ or $lb$ does not exist, will not be encoded and will remain a single variable.

The decision for each substitute variable can be obtained from the LNS policy (see Sec. 3.3), where 0 means the variable indicates reliability

---

**Algorithm 1** Bound tightening for Integer variable $x_i$

**Require:** Initial lower, upper bound of $x_i$: $lb$, $ub$;
  Current solution value: $x_i = p$;
  Binary LNS decision for $x_i$: $a_i^t$ for unbounded variables, and $\{a_{i,j}^t | j = 1, 2, ..., d\}$ for others.

**Ensure:** Tightened $lb$, $ub$

1: **if** $x_i$ unbounded **then**
2:     **if** $lb$ existed **and** $a_i^t = 0$ **then**
3:         $ub = 2p - lb$
4:     **else if** $ub$ existed **and** $a_i^t = 0$ **then**
5:         $lb = 2p - ub$
6:     **end if**
7: **else**
8:     $d = \lceil \log_2 (ub - lb) \rceil$
9:     **for** $j = 0 : d$ **do**
10:         **if** $a_{i,j}^t = 0$ **then**
11:             $lb = \max(lb, p - 1/2(ub - lb))$;
12:             $ub = \min(ub, p + 1/2(ub - lb))$;
13:         **else**
14:             break;
15:         **end if**
16:     **end for**
17: **end if**

---

at the current encoded bit, and 1 means it still needs exploration. We design a bound-tightening scheme to fully use the bit-wise decisions in Alg. 1 (see Appendix. A.9 for an example). Specifically, let $a_{i,j}^t$ represent the decision for the $j^{th}$ substitute variable of variable $i$ at step $t$. Decisions $a_{i,j}^t$ for all $j$ are checked, and the upper and lower bounds will be tightened around the current solution whenever $a_{i,j}^t = 0$, as in Line 11-12. Therefore, more fixed substitute variables can contribute to tighter bounds. In our embodiment, variables that sit far from both bounds can have a significantly wider exploration scope than close-to-bound variables, as they showed no explicit "preference" on either bound direction, which is significantly different from Nair et al. (2020b) (see Appendix A.1 for detailed discussion). Tightening on either bound when the current solution sits precisely at the midpoint of variable bounds, may contribute to performance degradation, which conceptually drives us to design the bound tightening scheme, tightening the bounds on the far side iteratively.

In addition, as for unbounded variables, meticulous analysis of MIPLIB2017 benchmark (Gleixner et al., 2021) revealed that all unbounded variables within the instances are characterized by unbounded in only one direction, which means that either $lb$ or $ub$ will exist for all general integer variables (otherwise it will be free to optimize in our implementation). In this respect, we define a virtual upper (lower) bound when $a_i^t = 0$ as in Line 3 and 5, which share similar insights with regular variables to put the current solution at precisely the midpoint of the updated bounds.

### 3.3 GRAPH-BASED LNS POLICY PARAMETERIZATION

A bipartite graph is recently popularly utilized in Gasse et al. (2019), Nair et al. (2020b), and Wu et al. (2021a) to represent the MIP instance states. However, the objective is not explicitly considered, which may contribute to performance degradation in some cases, e.g., when all discrete variables do not exist in the objectives (Yoon, 2022). To capture the correlations between objectives with variables and constraints reasonably, we propose to describe the input instance as a tripartite graph $\mathcal{G} = (\mathcal{V}, \mathcal{C}, \mathcal{O}, \mathcal{E})$, where $\mathcal{V}, \mathcal{C}$, and $\mathcal{O}$ denote the variable, constraint, and objective nodes, and $\mathcal{E}$ denotes the edges. The features of nodes and edges can refer to Appendix A.2, where the new objective node representations are defined as the average states of corresponding variables.

We parameterize the policy $\pi_\theta(a_t|s_t)$ by an attention-based Graph Convolution Network (GCN). Different from Graph Attention Networks (GATs) utilized in (Veličković et al., 2018; Ding et al., 2020), we remove the *softmax* normalization to fully reserve the raw weights between neighboring nodes and edges, capturing the contributions for each node to the final objectives (see Table 2, 4 for comparison with traditional message passing mechanism: **LNS-ATT**). The $\mathcal{C} \rightarrow \mathcal{V}$ passing is as follows (likewise for others):

$$\mathbf{h}_i^{t+1} = f_{\mathcal{C}\mathcal{V}} \left( \text{CONCAT} \left( \mathbf{h}_i^t, \frac{\sum\limits_{j \in \mathcal{C} \cap N_i} w_{ij}^t (\mathbf{h}_j^t + \mathbf{h}_{e_{ij}}^t)}{|\mathcal{C} \cap N_i|} \right) \right) \tag{3}$$

where $\mathbf{h}_i^t$ and $\mathbf{h}_{e_{ij}}^t$ are the features of node $i$ and edge $(i, j)$ at step $t$; $f_{\mathcal{C}\mathcal{V}}$ is a 2-layer perceptron with relu activation that maps the current states to the next iteration $\mathbf{h}_i^{t+1}$; $N_i$ denotes the neighborhood nodes of $i$ and $|\mathcal{C} \cap N_i|$ is the counts of neighborhood constraint nodes for node $i$, used to normalize the weighted sum neighboring features; $w_{ij}^t$ denotes the weighted coefficient between node $i$ and node $j$ at step $t$, measuring their correlations as follows, where $\mathbf{W}_{\mathcal{C}\mathcal{V}}$ is the weight matrix between constraint and variable.

$$w_{ij}^t = \sigma_s(\mathbf{W}_{\mathcal{C}\mathcal{V}} \cdot \text{CONCAT}(\mathbf{h}_i^t, \mathbf{h}_{e_{ij}}^t, \mathbf{h}_j^t)) \tag{4}$$

At each graph attention layer, the message passing between different types of nodes are $\mathcal{V} \rightarrow \mathcal{O}$, $\mathcal{O} \rightarrow \mathcal{C}, \mathcal{V} \rightarrow \mathcal{C}, \mathcal{C} \rightarrow \mathcal{O}, \mathcal{O} \rightarrow \mathcal{V}, \mathcal{C} \rightarrow \mathcal{V}$, which are calculated as Eq. 3 sequentially. In this way, after $K$ iterations, the features for both the nodes and edges are updated. We finally process the variable nodes by a multi-layer perceptron and the output value can be regarded as the *destroy* probability for each variable at this step, serving as the neighborhood search policy in Fig. 1. It is trained with Q-actor-critic by RL, following the same protocol with Wu et al. (2021a), while with the following differences:

**States**: We adopt an attentional tripartite graph to capture correlations among variables, constraints, and objectives. The features are gathered in Table 7 in the Appendix.

**Actions**: For the general variable $x_i$ with $d$ substitutes, the LNS decision at step $t$ will contain $d$ binary actions $a_{i,j}^t$, indicating the current solution reliable or not at each encoded bit $j$ (see Alg. 1).

**Transition and rewards**: We follow the same protocol as in (Wu et al., 2021a), where the next state $s_{t+1}$ is obtained by the baseline solver, and the reward is defined as objective improvements.

### 3.4 STEP-WISE GLOBAL INFORMATION BY BRANCHING

As discussed above, previous single-policy approaches were easily trapped in local optimum at an early stage in some complicated tasks, due to the learning complexity and limited exploration even in the training phase. To remedy this issue, an intuition is to select and optimize those wrongly fixed backdoor variables by LNS policy at each step. With this insight, we proposed to learn an extra branching network with

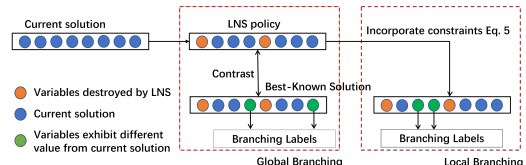

Figure 2: Global branching **vs** Local branching on different label collection schemes

imitation learning on top of LNS to filter out those variables at each step. Note that it was only applied to binary variables which are more likely to be backdoors that were fixed earlier, leading to local optima.

The most critical issue for the branching policy learning is the collection of branching variable labels. In other words, we need to figure out how to identify the potentially wrongly-fixed variables at each step. We proposed two different variants, which deal with the issue in global and local view respectively as in Fig. 2:

**Global branching (BTBS-LNS-G):** It gathers labels from the fixed variables by LNS at each step and contrasts them with the global optimal solution. Variables that exhibit differing values between these solutions are indicative of potentially misclassified variables within the current LNS decisions from a global perspective. Since the global optimal solution may be too difficult to acquire in a reasonable time, it was replaced by the best-known solution obtained across various approaches within the same time budget.

**Local branching (BTBS-LNS-L):** Different from the global view contrast, it gathers labels by incorporating the following local branching constraints (Liu et al., 2022) at each step:

$$\sum_{i \in \mathcal{B} \cap \mathcal{F}} |x_i^{t+1} - x_i^t| \le k \qquad (5)$$

where $\mathcal{F}$ is the currently fixed variables set by LNS. With this extra constraint, the re-defined sub-MIP can be solved by the baseline solver, and up to $K$ changed fixed variables will be selected at a local view as the branching variable labels at the current step. The selected variables are regarded as locally wrongly fixed variables by LNS.

With the collected labels for each variable, branch (1) or not (0), the branching network can be trained offline. The inputs are tripartite graph-based features (see Table 7 in the Appendix), where we additionally append the LNS decisions made by the learned LNS policy as variable features, as we only focused on the fixed variables for extra branching. Note that the input states are collected by resolving the training instances, along with the learned LNS policy. The labels are also gathered within the resolving at each step. Then the graph-based features are fed into a similar

---

**Algorithm 2** Offline training of branching policy for LNS

**Require:** graph-based states $S = \{s_t | t = 1, 2, ..., n\}$
  LNS decisions at each step $N = \{n_t | t = 1, 2, ..., n\}$
  branching variable labels $B = \{b_t | t = 1, 2, ..., n\}$ collected from the global or local branching;
**Ensure:** trained policy $\pi_\theta(B|S, N)$
 1: *// Samples are collected by resolving the training instances, along with the learned LNS;*
 2: Let $D = \{((s_t, n_t), b_t) | t = 1, 2, ..., n\}$.
 3: *// train the model;*
 4: Initialize all learnable parameters $\theta$;
 5: **while** stopping criteria not meet **do**
 6:    Randomly select a batch of instances $D_C$ from D;
 7:    Optimize $\theta$ by minimizing cross-entropy loss;
 8: **end while**

---

**Algorithm 3** Branch and search at the $t^{th}$ step

**Require:** Number of variables $n$;
  LNS decisions $N^t = \{n_i^t | i = 1, 2, ..., n\}$;
  branching decisions $B^t = \{b_i^t | i = 1, 2, ..., n\}$;
  variable set $\mathbf{x} = \{x_i | i = 1, 2, ..., n\}$;
  best solution at the $t^{th}$ step $\mathbf{x}^t = \{x_i^t | i = 1, 2, ..., n\}$;
  The ratio for branching variables $r$;
**Ensure:** $\mathbf{x}^{t+1}$;
 1: Let $D = \emptyset$;
 2: **for** $j = 0 : n$ **do**
 3:    **if** $x_i$ is general integer variable **then**
 4:       Tighten the bound as in Alg. 1 using $n_{i,j}^t$ (with $d$ separate decisions for each substitute variable);
 5:    **else**
 6:       **if** $b_i^t = 1$ and $x_i$ is binary variable **then**
 7:          $D = D \cup \{i\}$;
 8:       **else**
 9:          **if** $n_i^t = 0$ **then**
10:             Fix the value $x_i^{t+1} = x_i^t$;
11:          **end if**
12:       **end if**
13:    **end if**
14: **end for**
15: add constraint $\sum_{i \in D} |x_i^{t+1} - x_i^t| \le rn$ to sub-MIP;
16: Optimize $\mathbf{x}^{t+1}$ with the solver;

---

graph attention network as described in Sec. 3.3 to update the node/edge representations. We finally process the variable nodes by a multi-layer perceptron (MLP) and the output value can be regarded as the branching probability for each variable at this step. Cross-entropy loss was utilized to train the branching network to bring the outputs closer to the collected labels, with the pipeline as in Alg. 2.

Except for the label collection scheme, **BTBS-LNS-L** and **BTBS-LNS-G** remain all the same. The branching policy takes effect on top of LNS, enhancing exploration and optimizing its wrongly fixed backdoor variables at each step. The pipeline for the hybrid framework is given in Alg. 3, where

Table 2: Comparison on binary Integer Programming (IP) problems: SC, MIS, CA, MC. We also let SCIP run for a longer time (500s with SCIP (500s) and 1000s with SCIP (1000s), respectively). So for Gurobi and our **BTBS-LNS** in other tables.

| Methods | Set Covering (SC) | | Maximal Independent Set (MIS) | | Combinatorial Auction (CA) | | Maximum Cut (MC) | |
|---|---|---|---|---|---|---|---|---|
| | Gap% | PI | Gap% | PI | Gap% | PI (×10³) | Gap% | PI |
| SCIP | 3.23 | 20225 | 0.25 | 312.25 | 4.71 | 3312.4 | 8.01 | 15193 |
| SCIP (500s) | 1.40 | / | 0.18 | / | 3.36 | / | 7.11 | / |
| SCIP (1000s) | 1.06 | / | 0.09 | / | 2.40 | / | 6.87 | / |
| U-LNS | 3.84 | 22459 | 1.50 | 1145.4 | 9.42 | 4003.0 | 6.72 | 11565 |
| R-LNS | 4.17 | 23015 | 1.29 | 693.45 | 6.92 | 3631.2 | 6.33 | 10923 |
| FT-LNS | 3.48 | 20988 | 1.42 | 1103.7 | 9.83 | 4123.6 | 6.30 | 10554 |
| DINS | 3.97 | 22735 | 1.24 | 657.5 | 4.48 | 3337.4 | 5.75 | 10006 |
| GINS | 3.81 | 22197 | 0.75 | 683.6 | 6.90 | 3599.8 | 5.41 | 9765.0 |
| RINS | 3.63 | 21835 | 1.32 | 816.5 | 7.33 | 3843.4 | 6.04 | 10277 |
| RENS | 2.35 | 19112 | 0.79 | 792.36 | 4.40 | 3125.2 | 5.29 | 9116 |
| RL-LNS | 1.29 | 17623 | 0.07 | 182.63 | 2.36 | 2271.6 | 4.25 | 6538 |
| Branching | 1.72 | 18007 | 0.07 | 183.44 | 3.09 | 2492.7 | 3.99 | 6104 |
| GNN-GBDT | 1.78 | 18169 | 0.22 | 295.43 | 2.24 | 2206.9 | 4.85 | 7492 |
| CL-LNS | 0.92 | 17025 | 0.07 | 182.99 | 2.05 | 2198.5 | 3.03 | 3883.5 |
| LNS-TG | 0.66 | 16828 | 0.08 | 182.24 | 2.32 | 2247.8 | 3.05 | 4782.6 |
| LNS-Branch | 1.11 | 17234 | 0.09 | 182.19 | 2.36 | 2275.3 | 3.73 | 5840.0 |
| LNS-ATT | 0.65 | 16714 | 0.07 | 182.10 | 2.23 | 2231.5 | 2.99 | 3975.1 |
| **BTBS-LNS-L** | 0.47 | 16234 | 0.05 | 181.47 | 2.18 | 2196.8 | 1.99 | 2518 |
| **BTBS-LNS-G** | 0.35 | 16205 | 0.05 | 178.35 | 1.43 | 1998.9 | 0.59 | 785 |
| Gurobi | 0.75 | 16796 | 0 | 173.15 | 1.44 | 2075.4 | 0.62 | 842 |

Table 3: Generalization to large-scale binary IP instances using the trained policies from small problems in Sec. 4.2

| Methods | Set Covering (SC2) | | Maximal Independent Set (MIS2) | | Combinatorial Auction (CA2) | | Maximum Cut (MC2) | |
|---|---|---|---|---|---|---|---|---|
| | Gap% | PI | Gap% | PI | Gap% | PI (×10³) | Gap% | PI |
| SCIP | 4.51 | 14953 | 3.45 | 9542.1 | 17.87 | 12312 | 8.38 | 30039 |
| SCIP (500s) | 2.74 | / | 0.86 | / | 8.18 | / | 8.26 | / |
| SCIP (1000s) | 1.37 | / | 0.52 | / | 5.13 | / | 8.13 | / |
| U-LNS | 3.96 | 14268 | 0.97 | 2778.5 | 8.53 | 8032.5 | 7.03 | 24862 |
| R-LNS | 3.94 | 14392 | 0.71 | 2079.3 | 6.34 | 7050.0 | 6.52 | 22450 |
| FT-LNS | 4.49 | 14885 | 0.96 | 2765.6 | 9.08 | 8324.2 | 6.44 | 22347 |
| DINS | 2.99 | 13916 | 0.65 | 1935.4 | 6.11 | 6848.5 | 7.02 | 24815 |
| GINS | 3.14 | 14008 | 0.69 | 2011.5 | 6.74 | 7433.7 | 6.52 | 22477 |
| RINS | 2.95 | 13793 | 0.58 | 1844.7 | 6.55 | 7129.3 | 6.75 | 23619 |
| RENS | 2.78 | 13465 | 0.55 | 1782.6 | 6.02 | 6735.2 | 6.23 | 20959 |
| RL-LNS | 1.66 | 13007 | 0.51 | 1524.7 | 4.13 | 5933.4 | 3.20 | 8449.6 |
| Branching | 1.53 | 12916 | 0.55 | 1769.4 | 4.52 | 6142.7 | 3.19 | 7857.3 |
| GNN-GBDT | 1.78 | 13069 | 0.55 | 1549.3 | 3.44 | 5508.9 | 2.79 | 6533.7 |
| CL-LNS | 1.41 | 12914 | 0.41 | 1298.5 | 3.51 | 5621.7 | 2.83 | 7184.1 |
| **BTBS-LNS-L** | 0.51 | 12431 | 0.04 | 543.60 | 1.67 | 4800.3 | 1.45 | 3385.9 |
| **BTBS-LNS-G** | 0.68 | 12498 | 0.02 | 515.28 | 1.89 | 5012.6 | 1.44 | 3397.5 |
| Gurobi | 0.71 | 12528 | 0.01 | 495.88 | 3.60 | 5723.5 | 1.01 | 2195.6 |
| Methods | Set Covering (SC4) | | Maximal Independent Set (MIS4) | | Combinatorial Auction (CA4) | | Maximum Cut (MC4) | |
| | Gap% | PI | Gap% | PI | Gap% | PI (×10³) | Gap% | PI |
| SCIP | 5.41 | 15524 | 3.45 | 22745 | 16.61 | 25275 | 8.71 | 78510 |
| SCIP (500s) | 4.21 | / | 3.44 | / | 16.61 | / | 8.69 | / |
| SCIP (1000s) | 3.05 | / | 3.03 | / | 16.61 | / | 8.46 | / |
| U-LNS | 3.42 | 14814 | 1.41 | 9759.0 | 7.42 | 16470 | 7.39 | 68245 |
| R-LNS | 3.26 | 14747 | 0.98 | 7745.5 | 6.19 | 15875 | 6.98 | 64712 |
| FT-LNS | 3.75 | 14882 | 1.30 | 9150.3 | 8.30 | 17328 | 7.02 | 65329 |
| DINS | 3.23 | 14725 | 1.03 | 7982.4 | 5.02 | 14789 | 6.97 | 64593 |
| GINS | 3.28 | 14782 | 0.85 | 7244.7 | 5.99 | 15538 | 7.04 | 65778 |
| RINS | 2.96 | 14599 | 1.09 | 8218.0 | 5.78 | 15309 | 6.89 | 63575 |
| RENS | 2.95 | 14573 | 0.82 | 6972.1 | 5.17 | 14916 | 6.85 | 62998 |
| RL-LNS | 3.73 | 14866 | 0.57 | 5365.1 | 3.52 | 13572 | 3.76 | 39645 |
| Branching | 3.39 | 14689 | 0.64 | 5744.8 | 3.37 | 13349 | 4.21 | 42718 |
| GNN-GBDT | 3.45 | 14169 | 0.59 | 5233.5 | 2.86 | 12853 | 4.52 | 45423 |
| CL-LNS | 3.39 | 14325 | 0.45 | 4533.4 | 2.99 | 13025 | 3.29 | 37384 |
| **BTBS-LNS-L** | 0.84 | 13716 | 0.07 | 2140.4 | 1.39 | 11128 | 1.52 | 21195 |
| **BTBS-LNS-G** | 1.20 | 13789 | 0.11 | 2636.9 | 1.46 | 11705 | 1.51 | 20984 |
| Gurobi | 1.22 | 13795 | 0.04 | 2215.7 | 12.61 | 21959 | 5.38 | 51298 |

we fix or tighten the bounds for some variables by the LNS policy (see Line 4, 10), and select some variables that were labeled 1 by the branching policy (see Line 6, 7) for extra branching. The hybrid branch and search policy work together to formulate the sub-MIP at each step.

# 4 EXPERIMENTS

## 4.1 SETTINGS AND PROTOCOLS

**Peer methods.** We compare with the following baselines in a 200s time limit by default.

**1) SCIP (v7.0.3), Gurobi (v9.5.0)**: state-of-the-art open source and commercial solvers, and were fine-tuned with the aggressive mode to focus on improving the objectives.

**2) U-LNS(Wu et al., 2021a), R-LNS(Song et al., 2020)**: randomized LNS following its implementation as in Wu et al. (2021a) and Song et al. (2020).

**3) DINS (Ghosh, 2007), GINS (Maher et al., 2017), RINS (Danna et al., 2005) and RENS** (Berthold, 2014): heuristic-based LNS policies that were common utilized.

Figure 3: Performance on Item (Left) & AMIPLIB (Right).

**4) FT-LNS (Song et al., 2020), RL-LNS (Wu et al., 2021a), Branching (Sonnerat et al., 2021), CL-LNS (Huang et al., 2023b) and GNN-GBDT (Ye et al., 2023)**: some learning-based LNS policies with imitation learning or RL, following the same protocol as its original implementation.

**5) LNS-TG, LNS-Branch, LNS-IBT, LNS-IT, LNS-ATT**: Degraded versions of **BTBS-LNS**, where we i) replace the tripartite graph with bipartite graph (**LNS-TG**); ii) remove the extra branching (**LNS-Branch**); iii) remove the binarized encoding (**LNS-IBT**) and bound tightening (**LNS-IT**); iv) replace the attention-based graph network with widely used GAT (Veličković et al., 2018) (**LNS-ATT**). Refer to Appendix A.2 for details.

**6) BTBS-LNS-F**: A variant of **BTBS-LNS**, where we replace our bound tightening mechanism with (Nair et al., 2020b).

**Instances.** It covers both binary and MIP problems. We follow (Wu et al., 2021a) to test four NP-hard binary Integer Programming Problems: Set Covering (SC), Maximal Independent Set (MIS), Combinatorial Auction (CA), and Maximum Cut (MC). We generate 200, 20, and 100 instances as training, validation, and testing sets, respectively. To evaluate the generalization ability, we also generate scale-transfer test instances, such as SC2 and MIS4 in Table 3. The suffix number refers to instance scales, for which the details are gathered in Table 8 in Appendix A.2.

We also test on two MIP datasets in Machine Learning for Combinatorial Optimization (ML4CO) competition[3]: Balanced Item Placement (**Item**) and Anonymous MIPLIB (**AMIPLIB**), on their official testing instances. Balanced Item Placement contained 1050 binary variables, 33 continuous variables, and 195 constraints per instance. The anonymous MIPLIB consists of a curated set of instances from MIPLIB2017, a long-standing benchmark for MIP solvers with diverse distributions, in which general integer variables are included. We also show empirical results on the whole MIPLIB2017 benchmark set in Sec. 4.5, where our **BTBS-LNS** even surpasses Gurobi on average.

**Hyperparameters.** We run experiments on an Intel 2.50GHz CPU. Performance comparison on CPU **vs** GPU version of our approach is given in Appendix A.5. All the approaches were evaluated with three different seeds, and the average performance was reported (see detailed stability analysis in Appendix A.6). We use the open-source SCIP (v7.0.3) as the baseline solver by default (recall the blue box in Fig. 1). Gurobi version experiments are gathered in Appendix A.4. We train 20 epochs for each instance, with 50 iterations per epoch and a 2s re-optimization time limit per iteration. LNS and branching are trained sequentially, with RL (see Sec. 3.3) and imitation learning (see Sec. 3.4), respectively. The graph convolutional layers were set as $K = 2$ for both policies, with 64-dimensional latent representations for the nodes and edges. Specifically for branching, we set the max branching variables $k = 50$ in Eq. 5 for the local branching variant. In the inference phase, the branching variable ratio $r$ in Alg. 3 is empirically set to 10% for both branching variants. **BTBS-LNS by default denotes the local branching variant BTBS-LNS-L throughout this paper**.

**Evaluation metric.** We calculate the average primal gap (Nair et al., 2020b) to measure the gap between the current solution $\mathbf{x}$ and the best-known solution $\mathbf{x}^*$ found by all methods among the $N$ testing instances, within a fixed time limit $T_0$:

$$gap = \frac{1}{N} \sum_{i=1}^{N} \frac{|\mathbf{c}_i^\top \mathbf{x}_i - \mathbf{c}_i^\top \mathbf{x}_i^*|}{\max\{|\mathbf{c}_i^\top \mathbf{x}_i|, |\mathbf{c}_i^\top \mathbf{x}_i^*|\}} \tag{6}$$

We also calculate the average Primal Integral (PI, (Huang et al., 2023b; Achterberg et al., 2012)) to evaluate the anytime performance within the time limit:

$$PI = \frac{1}{N} \sum_{i=1}^{N} \left( \int_{t=0}^{T_0} \mathbf{c}_i^\top \mathbf{x}_i^t dt - T_0 \mathbf{c}_i^\top \mathbf{x}_i^* \right) \tag{7}$$

[3]https://www.ecole.ai/2021/ml4co-competition/

where $\mathbf{x}_i^t$ denotes the best solution within $t$ for instance $i$.

## 4.2 Overall Performance Evaluation

Table 2 compares the results for integer programming. As can be seen, compared with SCIP and all competing LNS baselines, both **BTBS-LNS-G** and **BTBS-LNS-L** achieve consistently superior performance across all problems. LNS-TG, LNS-Branch, and LNS-ATT are degraded versions of **BTBS-LNS**, and they all perform slightly worse, revealing the effectiveness of the attention-based tripartite graph and the extra branching policy. Comparing the two variants, **BTBS-LNS-G** delivers consistently superior performance over **BTBS-LNS-L**, and it even surpasses the leading commercial solver on SC, CA, and MC. **Note that detailed anytime performance on these instances are shown in Fig. 6 to Fig. 9 in Appendix A.7**, further revealing the effectiveness of **BTBS-LNS**.

We also test our method on two NP-hard MIP problems, and the results are gathered in Table 4. Note that the anytime primal gap comparison is also shown in Fig. 3. Our method consistently outperforms SCIP and the competing LNS baselines and is slightly worse than Gurobi, capable of finding even better solutions for around 27% test instances on both Item and AMIPLIB.

For the AMIPLIB problem, which contains a curated set of instances from MIPLIB, we split the instances into train, validation, and test sets by 70%, 15%, and 15% with cross-validation. Policies learned from diverse training instances are directly applied to the test set. We increase the solving and re-optimization time limit at each step to 1800s and 60s for the instances, as they are too large to be solved. Different from Wu et al. (2021a), we consistently utilize open-source SCIP as the baseline solver. As seen from Table 4 and Fig. 3, our method significantly outperforms SCIP and LNS baselines and even delivers slightly better performance than Gurobi at an early stage. LNS-IBT, LNS-IT and **BTBS-LNS-F** achieve significantly inferior performance than our **BTBS-LNS**, showing the effect of the "Binarized Tightening" technique and its superiority over Nair et al. (2020b).

Table 4: Performance on MIP instances.

| Methods | Item | | | AMIPLIB |
| | Obj | Gap% | PI | Gap% |
|---|---|---|---|---|
| SCIP | 23.33 | 50.73 | 4152.4 | 13.72 |
| SCIP (500s) | 19.83 | 39.41 | / | / |
| SCIP (1000s) | 17.02 | 31.05 | / | / |
| U-LNS | 20.39 | 44.29 | 3685.6 | 15.73 |
| R-LNS | 20.04 | 43.64 | 3485.0 | 14.96 |
| RL-LNS | 20.04 | 43.58 | 3498.5 | 12.55 |
| DINS | 18.08 | 37.23 | 3075.9 | 13.10 |
| GINS | 19.78 | 42.11 | 3514.7 | 13.64 |
| RINS | 20.53 | 44.88 | 3662.5 | 13.89 |
| RENS | 17.51 | 34.18 | 2925.0 | 11.75 |
| Branching | 18.84 | 40.12 | 3237.6 | 12.95 |
| LNS-TG | 18.05 | 37.85 | 3090.5 | 6.45 |
| LNS-Branch | 20.12 | 43.90 | 3537.0 | 9.32 |
| LNS-ATT | 15.54 | 26.91 | 2512.8 | 5.45 |
| LNS-IBT | / | / | / | 7.63 |
| LNS-IT | / | / | / | 7.65 |
| **BTBS-LNS-L** | 13.82 | 16.82 | 2030.3 | 4.19 |
| **BTBS-LNS-G** | 13.45 | 15.78 | 1912.5 | 4.35 |
| **BTBS-LNS-F** | / | / | / | 7.01 |
| Gurobi | **12.67** | **6.73** | **1895.6** | **0.81** |

## 4.3 Problem-scale Generalization Ability Study

We test the generalization ability in line with (Wu et al., 2021a) with a 200s time limit. We directly use the trained policies on small-scale problems in Sec. 4.2, with results shown in Table 3.

As can be seen, the two variants show similar performance on the generalized instances. Compared with SCIP and all the competing LNS baselines, our approach still delivers significantly superior performance, showing a better generalization ability. As the problem sizes become larger, it can produce even better results than Gurobi on SC2, SC4, CA2, CA4, and MC4, and only slightly inferior on the remaining 3 groups. It suggests that our policies can sometimes be more efficient for larger instances than the leading commercial solver. Notably, there is a significant gap between **BTBS-LNS** and Gurobi for Combinatorial Auction (CA), particularly on CA4.

Table 5: Evaluation on CA against Gurobi.

| Methods | CA2 | | CA4 | |
| | Obj | Gap% | Obj | Gap% |
|---|---|---|---|---|
| Gurobi | -218245 | 3.60 | -389396 | 12.61 |
| Gurobi(500s) | -224245 | 0.95 | -431626 | 3.14 |
| Gurobi(1000s) | **-225629** | **0.33** | -436188 | 2.11 |
| **BTBS-LNS** | -222590 | 1.67 | -439431 | 1.39 |
| **BTBS-LNS**(500s) | -225108 | 0.56 | **-445563** | **0** |

We further increase the time limit to 500s and 1000s respectively on CA, with results shown in Table 5. Our method consistently outperforms Gurobi with the same time limit. For CA4, it can even produce better solutions with a much shorter time limit. It empirically requires over 3 hours for Gurobi to deliver the same primal gap on CA4, being $58\times$ slower than our method.

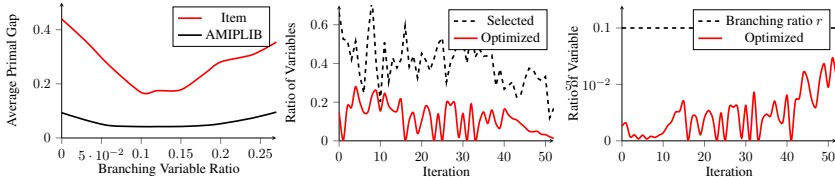

Figure 4: Impact of different branching ratios (Left). Selected & Optimized variables by the LNS (Middle) & Branching (Right) on Balanced Item Placement instances. The upper dotted line denotes the selected variable ratio that can be re-optimized by the learned Search & Branch policy, while the Solid red line denotes the variable ratio with solution value changes.

Table 6: Performance comparison on the whole MIPLIB2017 benchmark set.

|  | SCIP | SCIP(600s) | SCIP(900s) | U-LNS | R-LNS | FT-LNS | **BTBS-LNS** | **BTBS-LNS**-F | Gurobi |
|---|---|---|---|---|---|---|---|---|---|
| Gap% | 15.15 | 11.08 | 8.79 | 16.26 | 15.94 | 13.07 | **1.75** | 3.11 | 1.98 |

## 4.4 Branching Policy Study by Variable Ratios

To enhance exploration, an extra branching policy was developed to incorporate global information and help the learned LNS policy escape local optimum. Fig. 4 (**Left**) depicts the impact of branching variables ratios $r$ (see Alg. 3).

When the ratio $r < 0.1$, a larger size leads to better performance, optimizing some wrong decisions made by the learned LNS. Fig. 4 (**Right**) depicts the filtered and updated variable ratios. As can be seen, with increasing iterations, a growing number of LNS fixed variables were re-optimized by the additional branching policy, indicating the LNS decisions were sometimes unreliable. In other words, incorporating branching on top of LNS was essential to correct potential errors in the LNS decisions. However, when the branching size becomes extremely large, the performance significantly degrades constrained by the solving ability.

## 4.5 Experiments on MIPLIB2017 Benchmark

To further evaluate our proposed approach on some heterogeneous and hard instances, we also evaluated the whole MIPLIB2017 benchmark set from Hans Mittelmann. It contains 240 instances with diverse distributions and difficulties. We compared different methods in a 300s time limit, which is the geometric mean of solving time of the solved instances with SCIP, and the re-optimization time for each iteration was set as 5s. Other hyperparameters remain the same as **AMIPLIB** in Sec. 4.2. We perform **cross-validation** for a comprehensive comparison across all instances, splitting them into training, validation, and testing sets by 70%, 15%, and 15% respectively, at each round. The policies learned from the diverse training instances are then directly applied to the test set.

The overall comparison results were gathered in Table 6. As can be seen, our proposed **BTBS-LNS** can deliver 10% better primal gaps compared with Gurobi and achieve significantly better results compared with all the competing baselines. Compared with Gurobi, it can deliver better solutions on 12.4% instances, and obtained equally better solutions on 77% instances, indicating its effectiveness and generalization ability. Furthermore, we notice that **BTBS-LNS-F** performs slightly inferior to Gurobi and our approach, further revealing the superior performance of our Binarized Tightening technique over Nair et al. (2020b). **Detailed per-instance comparisons are gathered in Appendix A.8**. We also conducted specific experiments on unbounded variables from MIPLIB2017, which are given in Appendix A.3.

## 5 Conclusion and outlook

We have proposed a binarized tightening branch and search approach to learn LNS policies. It was designed to efficiently deal with general MIP problems and delivers superior performance over numerous competing baselines, including MIP solvers, learning and heuristic-based LNS approaches, on ILP, MIP datasets, and even heterogeneous instances from MIPLIB2017. Sufficient ablation studies demonstrate the effectiveness of each component. Considering the potential of our proposed **BTBS-LNS** on large-scale and cross-distribution instances, the applications in real-world scenarios may be our future direction.

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

## A  APPENDIX

### A.1  FURTHER DISCUSSION ON RELATED WORK

The main contributions of our **BTBS-LNS** are the general applicability and the addressing for local optima. Most LNS-based approaches (Liu et al., 2022; Nair et al., 2020a; Ding et al., 2020; Song et al., 2020; Wu et al., 2021a) solely deal with the binary programming problems due to their simplicity. Recently, some studies have tried to address the general MIP problems (Hendel, 2022; Sonnerat et al., 2021; Paulus et al., 2022a), in which Nair et al. (2020b) proposed a similar "bound tightening" technique. They differ from our approach in the following aspects. On one hand, the binary decision for each encoded variable was only applied for bound tightening in our approach, rather than directly fixed similar to Nair et al. (2020b). And on the other hand, the current solution value was also considered in bound tightening decisions in our approach. Variables that sit far from both bounds may have a significantly wider exploration scope than close-to-bound variables, as they showed no explicit "preference" in either direction. In addition, our approach can easily transfer to unbounded variables. We made a detailed comparison between the two approaches in Table 4 and Table 6. As can be seen, our **BTBS-LNS** consistently outperforms **BTBS-LNS-F**, demonstrating the effectiveness of our novel "Binarized Tightening" technique.

As for the local optima challenge, a few studies have tried Adaptive Neighborhood Size (ANS) (Huang et al., 2023a;b; Sonnerat et al., 2021) or hybrid heuristics control (Hendel, 2022), while it still requires hand-crafted hyperparameters, which are essential but difficult to determine. To address it more adaptively, we proposed to combine global-view information on top of LNS. When trapped in local optima, extra branching has the potential to select those wrongly fixed backdoor variables by the learned LNS policy for re-optimization. It is important to note that the concept of branching extends beyond the confines of the local branching (Sonnerat et al., 2021) and we also devised a novel variant termed "global branching", which can deliver even better performance in some cases. In addition, the major difference between our hybrid framework and the pure local branching approach (Sonnerat et al., 2021) lies in that we concentrate solely on variables fixed by LNS to correct its decisions, rather than the whole variable set. This specificity arises from the observation that LNS frequently converges to local optima when a limited number of backdoor variables are inaccurately fixed. Empirical results in Table 2, 3, and 4 demonstrated that our **BTBS-LNS** consistently outperforms the Branching baseline by Sonnerat et al. (2021).

We further review other studies related to ours, which can be divided into two categories: One is learning-based methods for specific MIP problems and the other is for general MIP problems.

**Policy learning for specific MIP Problems:** MIP problems cover numerous real-world tasks in many fields (Paschos, 2014) and quite a few studies attempt to solve certain types of problems, such as Traveling Salesman Problem (TSP) and Vehicle Routing Problems (VRP) (Li et al., 2021; Lu et al., 2019), etc. The algorithms can be divided into construction methods and learned improvement heuristics.

Construction methods usually attempt to directly learn approximate optimal solutions, like Graph Pointer Networks (GPNs) (Ma et al., 2019) and Monte Carlo tree search (Xing & Tu, 2020; Fu et al., 2021) for TSP instances. Compared to construction models, methods that learn improvement heuristics can often deliver better performance, by learning to iteratively improve the solution (Wu et al., 2021a). The improvement heuristics can be a guide for the next solution selection (Wu et al., 2021b), or policy to pick heuristics (Chen & Tian, 2019), or refinement from the current solution (Lu et al., 2019; Li et al., 2020). In general, both the learned improvement heuristics and construction methods have proved valid in some specific problems. In contrast, this paper aims to solve general MIP problems by learning improvement heuristic policies.

**Learning to solve general MIP problems:** Dual and primal are two main perspectives to improve solving efficiency for general MIP problems. Specifically, dual view aims to improve inner policies of Branch and Bound, like variable selection (Gasse et al., 2019; Zarpellon et al., 2021; Gupta et al., 2020), node selection (He et al., 2014) and cut selection (Tang et al., 2020; Paulus et al., 2022b;a). With a better decision at each node, the overall solving process can be greatly simplified.

In the primal perspective, the algorithms aim to find better feasible solutions by prediction or learning-based heuristics. For example, Ding et al. (2020) learned a tripartite graph-based deep

Table 7: Description of the tripartite graph features.

| Tensor | Feature Description |
|---|---|
| $\mathcal{V}$ | variable type (binary, integer, continuous). |
| | objective coefficient. |
| | lower and upper bound. |
| | reduced cost. |
| | solution value fractionality. |
| | **(dynamic)** solution value in incumbent. |
| | **(dynamic)** average solution value. |
| | **(dynamic)** best solution value. |
| | **(Branching Only)** LNS decisions at current step. |
| $\mathcal{C}$ | cosine similarity with objective. |
| | tightness indicator in LP solution. |
| | dual solution value. |
| | bias value, normalized with constraint coefficients |
| $\mathcal{O}$ | average states of related variables. |
| $\mathcal{V}$ - $\mathcal{C}$ | constraint coefficient per variable. |
| $\mathcal{V}$ - $\mathcal{O}$ | objective coefficient per variable. |
| $\mathcal{C}$ - $\mathcal{O}$ | constraint right-hand-side (RHS) coefficients. |

Table 8: Average variable/constraints of instances

| Num of | Training | | | | | | | Generalization | | | | |
|---|---|---|---|---|---|---|---|---|---|---|---|---|
| | SC | MIS | CA | MC | SC2 | MIS2 | CA2 | MC2 | SC4 | MIS4 | CA4 | MC4 |
| Variables | 1000 | 1500 | 4000 | 2975 | 2000 | 3000 | 8000 | 5975 | 4000 | 6000 | 16000 | 11975 |
| Constraints | 5000 | 5939 | 2674 | 4950 | 5000 | 11933 | 5344 | 9950 | 5000 | 23905 | 10717 | 19950 |

Table 9: Training, Validation and Test accuracy for graph-based branching network.

| | Local Branching | | | | | | Global Branching | | | | | |
|---|---|---|---|---|---|---|---|---|---|---|---|---|
| | SC | MIS | CA | MC | Item | AMIPLIB | SC | MIS | CA | MC | Item | AMIPLIB |
| Train% | 89.5 | 84.9 | 79.6 | 86.3 | 85.5 | 77.5 | 86.9 | 87.3 | 81.5 | 88.5 | 83.4 | 75.9 |
| Validation% | 84.8 | 83.5 | 75.1 | 82.1 | 82.8 | 74.9 | 83.7 | 84.9 | 80.9 | 87.0 | 81.8 | 75.1 |
| Test% | 82.5 | 81.6 | 72.9 | 80.5 | 81.5 | 74.2 | 83.1 | 82.6 | 80.1 | 84.5 | 80.7 | 73.8 |

neural network to generate partial assignments for binary variables, and to deal with the general integer variables, Nair et al. (2020b) proposed a bound tightening mechanism and learned partial assignments for each bit, respectively. Nevertheless, they were only applied in neural diving, and directly fixing may also lead to performance degradation or even infeasibility. To obtain broader applicability, learning-based primal heuristics, like large neighborhood search (Huang et al., 2023b; Song et al., 2020; Sonnerat et al., 2021; Nair et al., 2020a), and local branching (Liu et al., 2022), gradually catch more attention.

In this paper, we mainly focus on large neighborhood search heuristics, which have achieved remarkable progress in recent years. For example, Hendel (2022) designed an adaptive approach to combine multiple existing LNS heuristics to enhance the performance of a single policy, while it is largely limited by the rule-based heuristics and requires hand-crafted hyperparameters. To make it further, learning a better neighborhood function became popular recently. Sonnerat et al. (2021) and Song et al. (2020) both utilized imitation learning to select variable subsets to optimize at each step. However, the equal-size subsets make it inflexible and dramatically limit the performance. In this respect, Wu et al. (2021a) factorized the LNS policy into elementary actions on each variable and trained an RL-based policy to select variable subsets dynamically. However, current studies on LNS mainly focus on binary variables and are often susceptible to local optima even in the training phase due to the problem's complexity. In this respect, we propose a binarized-tightening branch and search approach to learn more efficient LNS policies for general problems.

## A.2 DETAIL FOR THE EXPERIMENTS

**Tripartite graph-based features:** We describe in Table 7 the variable, constraint, objective, and multi-source edge features of the tripartite graph utilized in both the LNS and branching policy

Table 10: Performance comparison on MIPLIB2017 instances that contained unbounded variables.

| Instance | SCIP | U-LNS | R-LNS | FT-LNS | BTBS-LNSw/o ubd | BTBS-LNS | Gurobi |
|---|---|---|---|---|---|---|---|
| gen-ip054 | 6858.879 | 6858.879 | 6852.733 | 6858.879 | 6852.733 | 6852.733 | 6840.966* |
| gen-ip002 | -4783.733* | -4772.597 | -4772.597 | -4768.253 | -4783.733* | -4783.733* | -4783.733* |
| **neos-3046615-murg** | 1610 | 1670 | 1651 | 1651 | 1610 | 1607 | 1600* |
| **buildingenergy** | 42652.34 | 42652.34 | 42652.34 | 42652.34 | 34243.89 | 33324.73 | 33283.85* |

learning in detail. Then we will clarify the features and connections for the encoded substitute variables. They are also characterized by static and dynamic features. For static features, the variable type and bounds are set to binary and 1/0, respectively, while other features are directly inherited from the original integer variables. For dynamic features, the solution value for each substitute variable is determined based on the encoded results. For example, for a general integer variable with a range of [0, 7], if the current solution value is 5, the solution values of the three substitute variables would be 1, 0, 1. Additionally, the connections between the substitute variables and other nodes are directly inherited from the original integer variables.

Except for the dynamic solving status, all the other features are collected at the root node of the search tree, and the dynamic features are collected along with the optimization process.

**Sizes for the generated instances with different difficulties:** The average variable and constraint size used in our experiments are listed in Table 8, which consists of small-scale training instances and some hard instances to evaluate the generalization ability.

**Accuracy for the imitation learning based branching policy:** Table 9 reports the training, validation, and testing accuracy of the global and local branching variants.

We compare our proposed **BTBS-LNS** with various baselines, which are explained as follows in detail:

**1) SCIP (v7.0.3), Gurobi (v9.5.0)**: state-of-the-art open source and commercial solver, and were fine-tuned with the aggressive mode to focus on improving the objectives.

**2) U-LNS**: an LNS version that uniformly samples variables at a fixed subset size. Note that for U-LNS, R-LNS and FT-LNS, we perform the same settings as Wu et al. (2021a).

**3) R-LNS**: an LNS version (Song et al., 2020) that randomly groups variables into equal subsets and re-optimizes them.

**4) DINS** (Ghosh, 2007), **GINS** (Maher et al., 2017), **RINS** (Danna et al., 2005) and **RENS** (Berthold, 2014): heuristic-based LNS policies.

**5) FT-LNS (Song et al., 2020)**: an LNS approach that applies imitation learning to learn the best R-LNS policies.

**6) RL-LNS (Wu et al., 2021a)**: Reinforcement learning LNS approach for variable subset optimization, while mainly focused on binary variable optimization.

**7) Branching (Sonnerat et al., 2021)**: An imitation learning-based LNS approach that learned from local branching constraints.

**8) LNS-TG**: A variant of our method, where we replace the tripartite graph with the widely used bipartite graph.

**10) LNS-Branch**: A variant of our method, where we remove the extra branching policy.

**11) LNS-IBT**: A variant of our method, where the general integer variables are equally treated as binary variables.

**12) LNS-IT**: A variant of our method, where we remove the "Tightening" technique and fix the integer variable to its current solution when either bit is fixed.

**13) LNS-ATT**: A variant of our method, where we replace our attention-based graph attention network with the widely used GAT.

**14) BTBS-LNS-F**: A variant of our method, where we replace our bound tightening mechanism with that proposed by Nair et al. (2020b).

Table 11: Experiments with Gurobi as the baseline for binary Integer Programming (IP)

| Methods | SC | | MIS | | CA | | MC | |
|---|---|---|---|---|---|---|---|---|
| | Gap% | PI | Gap% | PI | Gap% | PI($\times 10^3$) | Gap% | PI |
| U-LNS | 2.59 | 18820 | 0.41 | 635.32 | 3.78 | 2690.5 | 4.07 | 5633.8 |
| R-LNS | 3.01 | 18925 | 0.34 | 545.71 | 4.85 | 2999.0 | 3.75 | 5189.5 |
| FT-LNS | 3.38 | 19521 | 0.73 | 462.45 | 4.40 | 2856.4 | 3.79 | 5214.7 |
| RL-LNS | 1.57 | 16911 | 0.09 | 179.94 | 1.37 | 2029.1 | 3.52 | 4812.5 |
| **BTBS-LNS** | **0.28** | **15987** | **0** | **165.24** | **0.27** | **1710.6** | **0.38** | **426.89** |
| Gurobi | 0.75 | 16796 | **0** | 173.15 | 1.44 | 2075.4 | 0.62 | 842 |

Table 12: Generalization to large-scale binary integer programming (IP) instances with Gurobi as the baseline

| Methods | SC2 | | MIS2 | | CA2 | | MC2 | |
|---|---|---|---|---|---|---|---|---|
| | Gap% | PI | Gap% | PI | Gap% | PI($\times 10^3$) | Gap% | PI |
| U-LNS | 2.48 | 13599 | 0.32 | 1551.6 | 3.06 | 5442.1 | 3.76 | 13713 |
| R-LNS | 2.73 | 14052 | 0.55 | 1845.2 | 2.60 | 5112.0 | 3.75 | 13359 |
| FT-LNS | 3.29 | 14338 | 0.28 | 1485.2 | 3.72 | 5823.5 | 4.04 | 13753 |
| **BTBS-LNS** | **0.28** | **12275** | **0** | **462.38** | **0.47** | **4125.1** | **0.01** | **350.45** |
| Gurobi | 0.71 | 12528 | 0.01 | 495.88 | 3.60 | 5723.5 | 1.01 | 2195.6 |

| Methods | SC4 | | MIS4 | | CA4 | | MC4 | |
|---|---|---|---|---|---|---|---|---|
| | Gap% | PI | Gap% | PI | Gap% | PI($\times 10^3$) | Gap% | PI |
| U-LNS | 2.56 | 14150 | 0.64 | 5515.7 | 4.02 | 15712 | 4.95 | 46965 |
| R-LNS | 2.36 | 14112 | 0.52 | 4846.3 | 3.95 | 15275 | 4.84 | 46380 |
| FT-LNS | 3.34 | 14515 | 0.54 | 4915.0 | 3.68 | 14588 | 4.78 | 45795 |
| **BTBS-LNS** | **0.27** | **13424** | **0.01** | **2051.8** | **0.67** | **10025** | **0** | **11034** |
| Gurobi | 1.22 | 13795 | 0.04 | 2215.7 | 12.61 | 21959 | 5.38 | 51298 |

Note that the work by Sonnerat et al. (2021) doesn't have open-source code and some hyperparameters are difficult to fine-tune in different problems. However, to further evaluate our proposed framework with pure local branching based methods, we try to reproduce them, with the following details:

**1)** For a fair comparison, we replace the neural diving in Sonnerat et al. (2021) with an initial feasible solution generated by SCIP, the same as our approach.

**2)** In data collection, the desired Hamming radius $\eta_t$ are selected as 50, the same as our branching policy.

**3)** The model structure was the same as its descriptions, where we use the code provided by Gasse et al. (2019), and additionally use a fixed-size window (3 in the paper) of past variable assignments as variable features.

**4)** In the inference phase with the learned policy, we performed the same action sampling mechanism as in Sonnerat et al. (2021). As for the adaptive neighborhood size, we start with 10% of the integer variable size, and the dynamic factor $a$ was tuned from 1.01 to 1.05. Best-performing parameters will be selected for comparison in each problem. As a result, on SC and MIS, $a$ was set as 1.02, and $a = 1.03$ can deliver the best performance on other problems.

### A.3 DETAILED ANALYSIS ON MIPLIB2017

In Sec. 4.5, we evaluated our approach on the whole MIPLIB2017 benchmark set, which showed superior performance. To further evaluate the effectiveness of our novel virtual bound technique specifically for unbounded integer variables (see Alg. 1), we conducted an extensive analysis across all instances featured in the MIPLIB2017 benchmark set. Notably, there are 19 and 4 instances that contained unbounded integer variables before and after the

Table 13: Evaluation by Gurobi as baseline solver (MIP).

| Methods | | Item | | AMIPLIB |
|---|---|---|---|---|
| | Obj | Gap% | PI | Gap% |
| U-LNS | 17.64 | 36.08 | 3004.3 | 6.44 |
| R-LNS | 16.62 | 31.94 | 2788.6 | 6.01 |
| FT-LNS | 15.64 | 27.31 | 2519.4 | 5.45 |
| **BTBS-LNS** | **12.27** | **4.56** | **1823.7** | **0.47** |
| Gurobi | 12.67 | 6.73 | 1895.6 | 0.81 |

presolve, respectively. In this section, we compared our **BTBS-LNS** with a variant **BTBS-LNSw/o ubd**, where the special handling for unbounded integer variables (see Line 2-6 in Alg. 1) is removed. In other words, unbounded variables were free to optimize at each step. The comparison results on the four instances that still contain unbounded variables after presolve are gathered in Table 10.

Table 14: Average Standard Deviations for our proposed **BTBS-LNS** on different problems.

| Methods | SC | | SC2 | | SC4 | |
|---|---|---|---|---|---|---|
| | Obj | Gap% | Obj | Gap% | Obj | Gap% |
| **BTBS-LNS** | 547.88 ± 0.59% | 0.47 ± 0.88% | 293.56 ± 0.77% | 0.51 ± 0.68% | 169.80 ± 0.68% | 0.84 ± 1.01% |
| Methods | MIS | | MIS2 | | MIS4 | |
| | Obj | Gap% | Obj | Gap% | Obj | Gap% |
| **BTBS-LNS** | -685.86 ± 0.74% | 0.05 ± 0.78% | -1372.66 ± 0.51% | 0.04 ± 0.21% | -2747.04 ± 0.32% | 0.07 ± 0.19% |
| Methods | CA | | CA2 | | CA4 | |
| | Obj | Gap% | Obj | Gap% | Obj | Gap% |
| **BTBS-LNS** | -112864 ± 0.32% | 2.18 ± 0.29% | -222590 ± 0.39% | 1.67 ± 0.41% | -439431 ± 0.33% | 1.39 ± 0.49% |
| Methods | MC | | MC2 | | MC4 | |
| | Obj | Gap% | Obj | Gap% | Obj | Gap% |
| **BTBS-LNS** | -909.17 ± 0.48% | 1.99 ± 0.52% | -1831.00 ± 0.66% | 1.45 ± 0.58% | -3664 ± 0.73% | 1.52 ± 0.84% |
| Methods | Item | | AMIPLIB | | MIPLIB2017 | |
| | Obj | Gap% | Obj | Gap% | Obj | Gap% |
| **BTBS-LNS** | 13.82 ± 1.09% | 16.82 ± 0.96% | / | 4.19 ± 1.51% | / | 1.75 ± 1.62% |

As can be seen, our proposed **BTBS-LNS**, outperforms the variant **BTBS-LNSw/o ubd** on two instances and achieves parity on the other two. These findings underscore the potent effectiveness of our proposed bound-tightening technique, substantiating its value in enhancing solution quality and optimization efficiency. We will continue the experimentation on more unbounded MIP problems in the future.

### A.4 GUROBI VERSION OF OUR **BTBS-LNS**

To evaluate the performance of different approaches with Gurobi as the baseline solver, we perform extensive experiments on MIP problems, four binary integer programming problems, and their scale-transfer instances.

The hyperparameters remain unchanged from those in SCIP counterparts. The results of four binary integer programming problems and their scale-transfer instances are gathered in Table 11 and Table 12. The comparison results on MIP problems are reported in Table 13. As can be seen, our **BTBS-LNS** consistently outperforms Gurobi across all the problems with different sizes, indicating the effectiveness and generalization ability to different solvers.

### A.5 EXPERIMENTS WITH CPU VS GPU

All the experiments presented in Sec. 4 were performed on the Intel(R) Xeon(R) E5-2678 v3 2.50GHz CPU with 4 physical cores, and it achieved competitive performance even compared with the leading commercial solver. In this section, we will further evaluate the GPU version (NVIDIA GeForce RTX 2080) of our proposed **BTBS-LNS** on the balanced item placement problem.

Fig. 5 depicts the anytime primal gap comparison between the CPU and GPU versions in detail within the 200s time limit. As can be seen, compared with CPU implementation, GPU version **BTBS-LNS** delivers slightly better performance almost at any time, in which the overall primal gap and primal integral improve by 0.83% and 0.99%, respectively. In other words, our proposed **BTBS-LNS** may achieve even better performance when implemented in a GPU environment.

### A.6 STABILITY ANALYSIS OF OUR APPROACH

To make a fair comparison between different competing approaches, all the experiments in Sec. 4 were conducted with three different seeds. The average standard deviations for our proposed **BTBS-LNS** on different problems are gathered in Table 14. As can be seen, it is fairly robust to different seeds, with average standard deviations lower than 2% even on hard and heterogeneous problems, like MIPLIB2017.

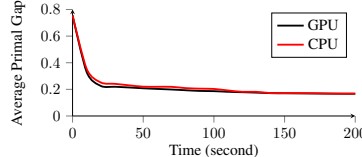

Figure 5: Anytime Performance comparison (GPU **vs** CPU).

### A.7 DETAILED ANYTIME PERFORMANCE ON INTEGER PROGRAMMING PROBLEMS

To further evaluate the anytime performance among the competing approaches, we plot the anytime primal gap curves on four binary integer programming problems, Set Covering (SC), Maximal Independent Set (MIS), Combinatorial Auction (CA), and Maximum Cut (MC), respectively. The results are gathered in Figure 6, 7, 8, 9, respectively.

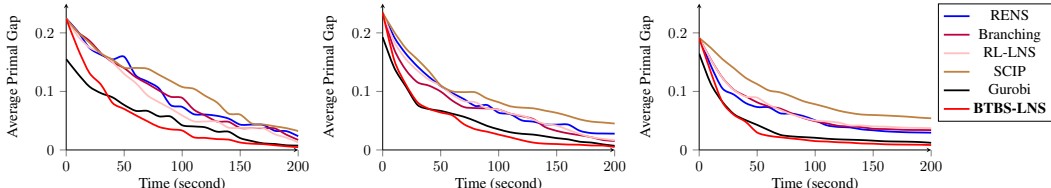

Figure 6: Anytime Performance on Set Covering (SC) problem and its scale-transfer instances. **From left to right**: Performance comparison on instances from SC, SC2, SC4. (see Table 8 for detail).

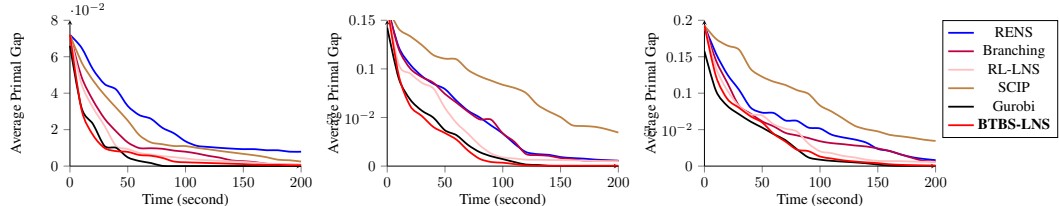

Figure 7: Anytime Performance on Maximal Independent Set (MIS) problem and its scale-transfer instances. **From left to right**: Performance comparison on instances from MIS, MIS2, MIS4. (see Table 8 for detail).

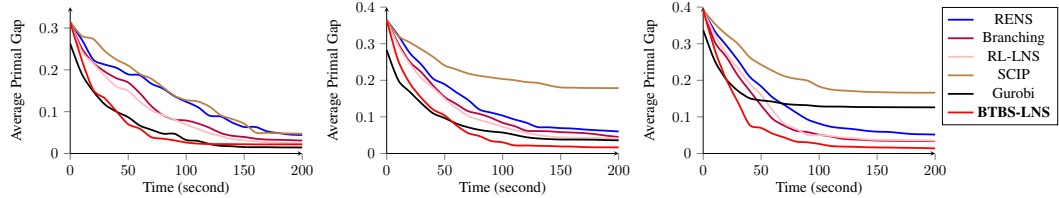

Figure 8: Anytime Performance on Combinatorial Auction (CA) problem and its scale-transfer instances. **From left to right**: Performance comparison on instances from CA, CA2, CA4. (see Table 8 for detail).

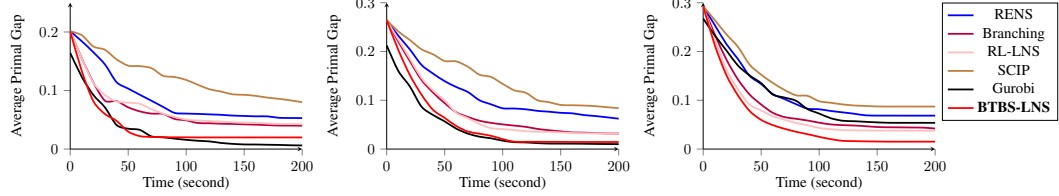

Figure 9: Anytime Performance on Maximum Cut (MC) problem and its scale-transfer instances. **From left to right**: Performance comparison on instances from MC, MC2, MC4. (see Table 8 for detail).

As seen from the results, our **BTBS-LNS** delivers consistently superior performance over the competing LNS baselines almost at any point, demonstrating its efficiency and effectiveness. More surprisingly, the proposed approach can achieve superior performance over the leading commercial solver in some cases, especially on the scale-transfer instances, purely by the learned policy on small-scale instances, with SCIP as the off-the-shelf solver.

## A.8   PER-INSTANCE PERFORMANCE COMPARISON ON MIPLIB2017

Considering that the results on MIPLIB2017 instances may deliver high variances due to the significantly different problem distributions across instances, showing only the average gap may not be sufficient. In this respect, we report the detailed per-instance performance within the given time limit on the competing approaches, and the results are gathered in Table 15.

We report **218**/240 instances from the MIPLIB2017 benchmark set. The following instances were removed, as no feasible solution can be found for them within the pre-defined time limit by the off-the-shelf solver SCIP:

**1) Instances that are infeasible (6)**:

- bnatt500
- cryptanalysiskb128n5obj14
- fhnw-binpack4-4
- neos-2075418-temuka
- neos-3988577-wolgan
- neos859080

**2) Instances that cannot generate feasible solution by the baseline solver within timelimit (16)**:

- cryptanalysiskb128n5obj16
- gfd-schedulen180f7d50m30k18
- highschool1-aigio
- irish-electricity
- neos-1354092
- neos-3402454-bohle
- neos-4532248-waihi
- neos-5104907-jarama
- neos-5114902-kasavu
- ns1116954
- ns1952667
- peg-solitaire-a3
- physiciansched3-3
- rail02
- supportcase19
- supportcase22

A.9   A SMALL NUMERICAL EXAMPLE FOR ALG. 1

Consider a general integer variable $x_0$ with a $[0, 15]$ range and a current solution of 3. This variable is encoded using 4 substitution binary variables: $x_{0,1}$, $x_{0,2}$, $x_{0,3}$, and $x_{0,4}$. At each LNS iteration $t$, distinct actions $a_{0,1}^t$, $a_{0,2}^t$, $a_{0,3}^t$, and $a_{0,4}^t$ are taken for each substitution variable, controlling the range of the original variable at different levels of significance. Specifically, the decision actions $a_{0,j}^t$ for all $j$ are evaluated sequentially, and the upper and lower bounds are tightened around the current solution whenever $a_{0,j}^t = 0$, as described in Line 11-12 of Alg. 1. Here are some examples:

1. If $a_{0,1}^t = 0$ and the others are 1, the updated bounds for $x_0$ will be $[0, 10]$.

2. If $a_{0,1}^t = a_{0,2}^t = 0$ and the others are 1, the updated bounds for $x_0$ will be $[0, 8]$.

3. ...

4. If all decision actions are 0, the updated bounds for $x_0$ will be $[0, 6]$, with the current solution precisely at the midpoint of the variable bounds. Similarly, when the current solution is 12, the updated bounds will be $[9, 15]$.

In each LNS iteration, our bound-tightening method refines the variable bounds around the current solution, guided by the LNS decisions, to balance computational complexity and exploration efficiency. Importantly, this bound-tightening process is conducted independently at each step, always initiating from the original variable bounds.

Table 15: Per-instance performance comparison on MIPLIB2017

| Instance | SCIP | SCIP(600s) | SCIP(900s) | U-LNS | R-LNS | FT-LNS | BTBS-LNS | BTBS-LNS-F | Gurobi | Optimal Solution |
|---|---|---|---|---|---|---|---|---|---|---|
| 30n20b8 | 302 | 302 | 302 | 353 | 302 | 302 | 302 | 302 | 302 | 302 |
| 50v-10 | 3340.37 | 3316.92 | 3313.18 | 3324.38 | 3340.37 | 3334.01 | 3311.18 | 3315.24 | 3311.18 | 3311.18 |
| academictimetablesmall | 228 | 228 | 228 | 228 | 228 | 228 | 6 | 45 | 6 | 0 |
| air05 | 26374 | 26374 | 26374 | 26439 | 26374 | 26441 | 26374 | 26374 | 26374 | 26374 |
| app1-1 | -3 | -3 | -3 | -3 | -2 | -3 | -3 | -3 | -3 | -3 |
| app1-2 | -23 | -41 | -41 | -24 | -23 | -29 | -41 | -41 | -41 | -41 |
| assign1-5-8 | 212 | 212 | 212 | 214 | 212 | 212 | 212 | 212 | 212 | 212 |
| atlanta-ip | 98.01 | 93.01 | 93.01 | 93.01 | 95.01 | 93.01 | 90.01 | 90.01 | 90.01 | 90.01 |
| b1c1s1 | 27031.16 | 27004.55 | 25571.02 | 27540.75 | 27031.16 | 26379.4 | 24544.25 | 24544.25 | 24544.25 | 24544.25 |
| bab2 | -354064.7 | -354064.7 | -354064.7 | -354064.7 | -354064.7 | -354091 | -354092.9 | -354092.9 | -357525.96 | -357544.312 |
| bab6 | -279121.2 | -279121.2 | -2791212 | -280546.4 | -280546.4 | -280546.4 | -280546.4 | -280546.4 | -284248.23 | -284248.23 |
| beasleyC3 | 754 | 754 | 754 | 759 | 759 | 755 | 754 | 754 | 754 | 754 |
| binkar10.1 | 6742.2 | 6742.2 | 6742.2 | 6746.76 | 6747.78 | 6743.24 | 6742.2 | 6742.2 | 6742.2 | 6742.2 |
| blp-ar98 | 6565.99 | 6303.11 | 6243.77 | 6565.99 | 6584.43 | 6565.99 | 6205.21 | 6205.21 | 6205.21 | 6205.21 |
| blp-ic98 | 4744.08 | 4719.1 | 4641.77 | 4719.1 | 4963.66 | 4963.66 | 4491.45 | 4491.45 | 4491.45 | 4491.45 |
| bnatt400 | 1 | 1 | 1 | 1 | 1 | 1 | 1 | 1 | 1 | 1 |
| bppc4-08 | 53 | 53 | 53 | 56 | 56 | 54 | 53 | 53 | 53 | 53 |
| brazil3 | 102 | 102 | 102 | 102 | 102 | 102 | 24 | 41 | 24 | 24 |
| buildingenergy | 42652.34 | 34250.38 | 34250.38 | 42652.34 | 42652.34 | 42652.34 | 33324.73 | 34250.38 | 33283.85 | 33283.85 |
| cbs-cta | 0 | 0 | 0 | 43.16 | 43.16 | 0 | 0 | 0 | 0 | 0 |
| chromaticindex1024-7 | 4 | 4 | 4 | 4 | 4 | 4 | 4 | 4 | 4 | 4 |
| chromaticindex512-7 | 4 | 4 | 4 | 4 | 4 | 4 | 4 | 4 | 4 | 4 |
| cmflsp50-24-8-8 | 57921400 | 57921400 | 57921400 | 57921400 | 57921400 | 57921400 | 55789390 | 55789390 | 55789390 | 55789390 |
| CMS750_4 | 261 | 254 | 252 | 261 | 269 | 253 | 252 | 252 | 252 | 252 |
| co-100 | 118333720 | 118333720 | 118333720 | 118833720 | 118833720 | 118333720 | 2639942.06 | 2639942.06 | 2639942.06 | 2639942.06 |
| cod105 | -12 | -12 | -12 | -11 | -11 | -8 | -12 | -12 | -12 | -12 |
| comp07-2idx | 823 | 148 | 148 | 148 | 148 | 78 | 6 | 23 | 6 | 6 |
| comp21-2idx | 250 | 179 | 142 | 250 | 250 | 225 | 75 | 88 | 88 | 74 |
| cost266-UUE | 25222800 | 25148941 | 25148941 | 25222800 | 25222800 | 25164070 | 25148941 | 25148941 | 25148941 | 25148941 |
| csched007 | 362 | 351 | 351 | 362 | 356 | 354 | 351 | 351 | 351 | 351 |
| csched008 | 173 | 173 | 173 | 176 | 178 | 174 | 173 | 173 | 173 | 173 |
| cvs16r128-89 | -93 | -95 | -95 | -86 | -80 | -84 | -97 | -97 | -96 | -97 |
| dano3_3 | 576.345 | 576.345 | 576.345 | 577.475 | 576.52 | 576.52 | 576.345 | 576.345 | 576.345 | 576.345 |
| dano3_5 | 576.925 | 576.925 | 576.925 | 581.725 | 581.725 | 577.316 | 576.925 | 576.925 | 576.925 | 576.925 |
| decomp2 | -160 | -160 | -160 | -152 | -152 | -133 | -160 | -160 | -160 | -160 |
| drayage-100-23 | 103334 | 103334 | 103334 | 103334 | 103334 | 103334 | 103334 | 103334 | 103334 | 103333.874 |
| drayage-25-23 | 101283 | 101283 | 101283 | 106897 | 101344 | 101344 | 101283 | 101283 | 101283 | 101282.647 |
| dws008-01 | 56691.23 | 46179.85 | 38873.46 | 56691.23 | 56691.23 | 56691.23 | 37412.6 | 37412.6 | 37412.6 | 37412.6 |

| Instance | SCIP | SCIP(600s) | SCIP(900s) | U-LNS | R-LNS | FT-LNS | BTBS-LNS | BTBS-LNS-F | Gurobi | Optimal Solution |
|---|---|---|---|---|---|---|---|---|---|---|
| eil33-2 | 934.008 | 934.008 | 934.008 | 987.674 | 987.674 | 934.008 | 934.008 | 934.008 | 934.008 | 934.008 |
| eilA101-2 | 1313.47 | 995.77 | 995.77 | 1443.53 | 1443.53 | 1443.53 | 880.92 | 880.92 | 923.01 | 880.92 |
| enlight_hard | 37 | 37 | 37 | 37 | 37 | 37 | 37 | 37 | 37 | 37 |
| ex10 | 100 | 100 | 100 | 100 | 100 | 100 | 100 | 100 | 100 | 100 |
| ex9 | 81 | 81 | 81 | 81 | 81 | 81 | 81 | 81 | 81 | 81 |
| exp-1-500-5-5 | 65887 | 65887 | 65887 | 65887 | 65887 | 65887 | 65887 | 65887 | 65887 | 65887 |
| fast0507 | 174 | 174 | 174 | 176 | 175 | 174 | 174 | 174 | 174 | 174 |
| fastxgemm-n2r6s0t2 | 230 | 230 | 230 | 230 | 236 | 230 | 230 | 230 | 230 | 230 |
| fhnw-binpack4-48 | 0 | 0 | 0 | 0 | 0 | 0 | 0 | 0 | 0 | 0 |
| fiball | 140 | 138 | 138 | 138 | 140 | 138 | 138 | 138 | 138 | 138 |
| gen-ip002 | -4783.73 | -4783.73 | -4783.73 | -4772.6 | -4772.6 | -4768.25 | -4783.73 | -4772.33 | -4783.73 | -4783.73 |
| gen-ip054 | 6858.88 | 6840.97 | 6840.97 | 6858.88 | 6852.73 | 6858.88 | 6852.73 | 6858.58 | 6840.97 | 6840.97 |
| germanr | 48440630 | 48096190 | 48096190 | 48440630 | 48440630 | 48440630 | 48440630 | 48440630 | 47135500 | 47095869.6 |
| glass-sc | 23 | 23 | 23 | 25 | 25 | 24 | 23 | 23 | 23 | 23 |
| glass4 | 1200012600 | 1200012600 | 1200012600 | 1200012600 | 1200012600 | 1200012600 | 1200012600 | 1200012600 | 1200012600 | 1200012600 |
| gmu-35-40 | -2406458 | -2406458 | -2406458 | -2406458 | -2406458 | -2406458 | -2406733 | -2406458 | -2406733 | -2406733.37 |
| gmu-35-50 | -2606871 | -2606930 | -2606930 | -2605465 | -2605387 | -2606930 | -2607958.3 | -2607958.3 | -2607922.7 | -2607958.33 |
| graph20-20-1rand | -9 | -9 | -9 | -8 | -8 | -9 | -9 | -9 | -9 | -9 |
| graphdraw-domain | 19686 | 19686 | 19686 | 19848 | 19848 | 19772 | 19686 | 19688 | 19686 | 19686 |
| h80x6320d | 6382.1 | 6382.1 | 6382.1 | 6382.1 | 6382.1 | 6382.1 | 6382.1 | 6382.1 | 6382.1 | 6382.1 |
| hypothyroid-k1 | -2851 | -2851 | -2851 | -2851 | -2851 | -2851 | -2851 | -2851 | -2851 | -2851 |
| ic97_potential | 3945 | 3945 | 3945 | 3952 | 3945 | 3952 | 3942 | 3952 | 3942 | 3942 |
| icir97_tension | 6392 | 6382 | 6375 | 6375 | 6382 | 6376 | 6375 | 6375 | 6375 | 6375 |
| irp | 12159.49 | 12159.49 | 12159.49 | 12160.2 | 12161.5 | 12161.5 | 12159.49 | 12159.49 | 12159.49 | 12159.49 |
| istanbul-no-cutoff | 204.08 | 204.08 | 204.08 | 214.797 | 212.961 | 214.797 | 204.08 | 204.08 | 204.08 | 204.08 |
| k1mushroom | -204 | -293 | -3288 | -204 | -204 | -204 | -3144 | -3144 | -3288 | -3288 |
| lectsched-5-obj | 48 | 41 | 39 | 46 | 48 | 44 | 24 | 27 | 24 | 24 |
| leo1 | 419655200 | 412600400 | 410709200 | 488216000 | 443395100 | 429182100 | 404989400 | 404989400 | 404227536 | 404227536 |
| leo2 | 436700200 | 426090600 | 424958900 | 438115100 | 436700200 | 436444000 | 405531200 | 405531200 | 404077441 | 404077441 |
| lotsize | 1557868 | 1484323 | 1483960 | 1626587 | 1557868 | 1495682 | 1480195 | 1480195 | 1494101 | 1480195 |
| mad | 0.067 | 0.038 | 0.0352 | 0.067 | 0.0772 | 0.0392 | 0.0268 | 0.0268 | 0.028 | 0.0268 |
| map10 | -480 | -495 | -495 | -468 | -410 | -472 | -495 | -495 | -495 | -495 |
| map16715-04 | -78 | -109 | -111 | -82 | -78 | -83 | -111 | -111 | -111 | -111 |
| markshare_4_0 | 1 | 1 | 1 | 3 | 1 | 1 | 1 | 1 | 1 | 1 |
| markshare2 | 31 | 28 | 28 | 31 | 36 | 31 | 24 | 24 | 24 | 24 |
| mas74 | 11801.19 | 11801.19 | 11801.19 | 11801.19 | 11801.19 | 11801.19 | 11801.19 | 11801.19 | 11801.19 | 11801.1857 |
| mas76 | 40005.05 | 40005.05 | 40005.05 | 40005.05 | 40005.05 | 40005.05 | 40005.05 | 40005.05 | 40005.05 | 40005.05 |
| mc11 | 11689 | 11689 | 11689 | 11720 | 11731 | 11896 | 11689 | 11689 | 11689 | 11689 |
| mcsched | 211913 | 211913 | 211913 | 212874 | 212874 | 212911 | 211913 | 211913 | 211913 | 211913 |
| mik-250-20-75-4 | -52301 | -52301 | -52301 | -52301 | -52301 | -52301 | -52301 | -52301 | -52301 | -52301 |
| milo-v12-6-r2-40-1 | 326481.1 | 326481.1 | 326481.1 | 326820.6 | 326820.6 | 326481.1 | 326481.1 | 326481.1 | 326481.1 | 326481.1 |
| momentum1 | 372399.4 | 282447.1 | 134897 | 365944 | 365944 | 372399.4 | 109143.5 | 109143.5 | 109143.5 | 109143.5 |
| mushroom-best | 0.0553 | 0.0553 | 0.0553 | 0.0869 | 0.0869 | 0.0553 | 0.0553 | 0.0553 | 0.0553 | 0.0553 |
| mzzv11 | -21718 | -21718 | -21718 | -21678 | -21668 | -21678 | -21718 | -21718 | -21718 | -21718 |
| mzzv42z | -20540 | -20540 | -20540 | -20540 | -20540 | -20400 | -20540 | -20540 | -20540 | -20540 |

| Instance | SCIP | SCIP(600s) | SCIP(900s) | U-LNS | R-LNS | FT-LNS | BTBS-LNS | BTBS-LNS-F | Gurobi | Optimal Solution |
|---|---|---|---|---|---|---|---|---|---|---|
| n2seq36q | 52600 | 52200 | 52200 | 52800 | 52600 | 52400 | 52200 | 52200 | 52200 | 52200 |
| n3div36 | 130800 | 130800 | 130800 | 130800 | 131400 | 130800 | 130800 | 130800 | 130800 | 130800 |
| n5-3 | 8105 | 8105 | 8105 | 8405 | 8105 | 8105 | 8105 | 8105 | 8105 | 8105 |
| neos-1122047 | 161 | 161 | 161 | 161 | 161 | 161 | 161 | 161 | 161 | 161 |
| neos-1171448 | -309 | -309 | -309 | -307 | -305 | -309 | -309 | -309 | -309 | -309 |
| neos-1171737 | -190 | -192 | -192 | -173 | -190 | -190 | -195 | -195 | -195 | -195 |
| neos-1445765 | -17783 | -17783 | -17783 | -17783 | -17783 | -17783 | -17783 | -17783 | -17783 | -17783 |
| neos-1456979 | 186 | 184 | 184 | 207 | 184 | 186 | 176 | 178 | 176 | 176 |
| neos-1582420 | 91 | 91 | 91 | 91 | 91 | 91 | 91 | 91 | 91 | 91 |
| neos-2657525-crna | 7.23 | 7.23 | 7.23 | 7.23 | 8.06 | 7.23 | 1.81075 | 7.23 | 1.81075 | 1.81075 |
| neos-2746589-doon | 2099.6 | 2099.6 | 2099.6 | 2099.6 | 2099.6 | 2099.6 | 2008.2 | 2099.6 | 2008.2 | 2008.2 |
| neos-2978193-inde | -2.388 | -2.388 | -2.388 | -2.197 | -2.388 | -2.388 | -2.388 | -2.388 | -2.388 | -2.38806169 |
| neos-2987310-joes | -607702988 | -607702988 | -607702988 | -607702988 | -607702988 | -607702988 | -607702988 | -607702988 | -607702988 | -607702988 |
| neos-3004026-krka | 0 | 0 | 0 | 0 | 0 | 0 | 0 | 0 | 0 | 0 |
| neos-3024952-loue | 126520 | 97446 | 71336 | 81469 | 97446 | 126520 | 26756 | 27349 | 26756 | 26756 |
| neos-3046615-murg | 1610 | 1610 | 1607 | 1670 | 1651 | 1651 | 1607 | 1611 | 1600 | 1600 |
| neos-3083819-nubu | 6307996 | 6307996 | 6307996 | 6307996 | 6307996 | 6307996 | 6307996 | 6307996 | 6307996 | 6307996 |
| neos-3216931-puriri | 151160 | 151160 | 151160 | 141275 | 151160 | 141275 | 141275 | 141275 | 71320 | 71320 |
| neos-3381206-awhea | 453 | 453 | 453 | 453 | 454 | 453 | 453 | 454 | 453 | 453 |
| neos-3402294-bobin | 0.06725 | 0.06725 | 0.06725 | 0.08775 | 0.06725 | 0.08175 | 0.06725 | 0.06725 | 0.06725 | 0.06725 |
| neos-3555904-turama | -34.7 | -34.7 | -34.7 | -34.7 | -34.7 | -34.7 | -34.7 | -34.7 | -34.7 | -34.7 |
| neos-3627168-kasai | 989301.6 | 989301.6 | 989301.6 | 990006.8 | 989301.6 | 989301.6 | 988585.62 | 988585.62 | 988585.62 | 988585.62 |
| neos-3656078-kumeu | -11067.1 | -11067.1 | -11067.1 | -11067.1 | -11067.1 | -11067.1 | -13127 | -13120 | -13171 | -13171 |
| neos-3754480-nidda | 13832.17 | 13639.97 | 13639.97 | 13832.17 | 13832.17 | 13832.17 | 12940.5 | 12940.5 | 12941.69 | 12940.5 |
| neos-4300652-rahue | 7.4454 | 2.8193 | 2.7595 | 6.1813 | 7.4454 | 6.1813 | 2.1416 | 2.1416 | 2.1416 | 2.1416 |
| neos-4338804-snowy | 1477 | 1474 | 1473 | 1482 | 1479 | 1479 | 1471 | 1473 | 1471 | 1471 |
| neos-4387871-tavua | 35.14 | 35.14 | 35.14 | 35.14 | 35.14 | 35.14 | 33.38 | 33.38 | 33.38 | 33.38 |
| neos-4413714-turia | 45.37 | 45.37 | 45.37 | 51.94 | 51.94 | 45.37 | 45.37 | 45.37 | 45.37 | 45.37 |
| neos-4647030-tutaki | 27268.48 | 27268.48 | 27268.48 | 27268.48 | 27268.48 | 27268.48 | 27265.71 | 27265.71 | 27265.71 | 27265.71 |
| neos-4722843-widden | 25438.44 | 25210.88 | 25210.88 | 27707.88 | 26277.44 | 26277.44 | 25009.7 | 25309.66 | 25009.7 | 25009.7 |
| neos-4738912-atrato | 285010500 | 283680800 | 283680100 | 285662900 | 285662900 | 285010500 | 283627957 | 283627957 | 283627957 | 283627957 |
| neos-4763324-toguru | 6760.735 | 6760.735 | 6760.735 | 6760.735 | 6760.735 | 6760.735 | 1613.039 | 1613.039 | 1613.039 | 1613.039 |
| neos-4954672-berkel | 2678506 | 2627560 | 2624735 | 2678506 | 2678506 | 2678506 | 2612710 | 2612710 | 2614881 | 2612710 |
| neos-5049753-cuanza | 636 | 636 | 636 | 636 | 636 | 600 | 562 | 600 | 562 | 562 |
| neos-5052403-cygnet | 293 | 293 | 184 | 293 | 293 | 293 | 182 | 182 | 182 | 182 |
| neos-5093327-huahum | 6686 | 6686 | 6686 | 6686 | 6960 | 6686 | 6260 | 6260 | 6270 | 6260 |
| neos-5107597-kakapo | 4248 | 3744 | 3654 | 4194 | 4293 | 4158 | 3645 | 3645 | 3645 | 3645 |
| neos-5188808-nattai | 0.11257 | 0.11257 | 0.11207 | 0.11257 | 0.11257 | 0.11257 | 0.11029 | 0.11029 | 0.11029 | 0.11029 |
| neos-5195221-niemur | 0.00406 | 0.00384 | 0.00384 | 0.00418 | 0.00418 | 0.00406 | 0.00384 | 0.00384 | 0.00384 | 0.00384 |
| neos-631710 | 214 | 214 | 214 | 214 | 214 | 214 | 203 | 203 | 203 | 203 |
| neos-662469 | 245034.5 | 184745.5 | 184679.5 | 224993.5 | 225044 | 245034.5 | 184380 | 184390 | 184380 | 184380 |
| neos-787933 | 30 | 30 | 30 | 30 | 30 | 30 | 30 | 30 | 30 | 30 |
| neos-827175 | 112.002 | 112.002 | 112.002 | 112.002 | 112.002 | 112.002 | 112.002 | 112.002 | 112.002 | 112.002 |
| neos-848589 | 12359660 | 2359.54 | 2359.54 | 12359660 | 12359660 | 12359660 | 2358.43 | 2358.43 | 3206.12 | 2358.43 |
| neos-860300 | 3201 | 3201 | 3201 | 3201 | 3267 | 3201 | 3201 | 3201 | 3201 | 3201 |

| Instance | SCIP | SCIP(600s) | SCIP(900s) | U-LNS | R-LNS | FT-LNS | BTBS-LNS | BTBS-LNS-F | Gurobi | Optimal Solution |
|---|---|---|---|---|---|---|---|---|---|---|
| neos-873061 | 122.92 | 122.72 | 122.72 | 125.93 | 122.92 | 123.66 | 113.656 | 113.656 | 113.656 | 113.656 |
| neos-911970 | 54.76 | 54.76 | 54.76 | 54.83 | 54.83 | 54.76 | 54.76 | 54.76 | 54.76 | 54.76 |
| neos-933966 | 2388 | 320 | 320 | 2389 | 2388 | 2388 | 318 | 318 | 318 | 318 |
| neos-950242 | 4 | 4 | 4 | 4 | 5 | 4 | 4 | 4 | 4 | 4 |
| neos-957323 | -237.76 | -237.76 | -237.76 | -234.76 | -235.76 | -235.76 | -237.76 | -237.76 | -237.76 | -237.76 |
| neos-960392 | 0 | -238 | -238 | 0 | 0 | -234 | -238 | -238 | -238 | -238 |
| neos17 | 0.15 | 0.15 | 0.15 | 0.171 | 0.167 | 0.151 | 0.15 | 0.15 | 0.15 | 0.15 |
| neos5 | 15 | 15 | 15 | 15 | 15 | 15 | 15 | 15 | 15 | 15 |
| neos8 | -3719 | -3719 | -3719 | -3719 | -3719 | -3719 | -3719 | -3719 | -3719 | -3719 |
| net12 | 214 | 214 | 214 | 214 | 255 | 214 | 214 | 214 | 214 | 214 |
| netdiversion | 4900438 | 4900438 | 263 | 4900438 | 4900438 | 263 | 242 | 244 | 242 | 242 |
| nexp-150-20-8-5 | 300 | 234 | 231 | 771 | 771 | 237 | 231 | 231 | 239 | 231 |
| ns1208400 | 2 | 2 | 2 | 2 | 2 | 2 | 2 | 2 | 2 | 2 |
| ns1644855 | -1419.67 | -1524.33 | -1524.33 | -1486.67 | -1486.67 | -1419.67 | -1524.33 | -1524.33 | -1524.33 | -1524.33 |
| ns1760995 | -429.36 | -429.36 | -429.36 | -429.36 | -429.36 | -429.36 | -548.02 | -548.02 | -516.07 | -549.214385 |
| ns1830653 | 20622 | 20622 | 20622 | 23622 | 23622 | 21622 | 20622 | 20622 | 20622 | 20622 |
| nu25-pr12 | 53905 | 53905 | 53905 | 53905 | 53905 | 53905 | 53905 | 53905 | 53905 | 53905 |
| nursesched-medium-hint03 | 8080 | 8080 | 7906 | 8080 | 8080 | 8080 | 117 | 997 | 152 | 115 |
| nursesched-sprint02 | 58 | 58 | 58 | 58 | 67 | 58 | 58 | 58 | 58 | 58 |
| nw04 | 16862 | 16862 | 16862 | 16876 | 16876 | 16876 | 16862 | 16862 | 16862 | 16862 |
| opm2-z10-s4 | -29112 | -33062 | -33062 | -26538 | -26538 | -26538 | -33269 | -33269 | -33139 | -33269 |
| p200x1188c | 15078 | 15078 | 15078 | 15078 | 15078 | 15078 | 15078 | 15078 | 15078 | 15078 |
| pg | -8674.34 | -8674.34 | -8674.34 | -8662.84 | -8662.84 | -8674.34 | -8674.34 | -8674.34 | -8674.34 | -8674.34 |
| pg5_34 | -14324.46 | -14324.81 | -14325.83 | -14310.96 | -14324.46 | -14324.81 | -14339.4 | -14339.4 | -14339.4 | -14339.4 |
| physiciansched6-2 | 49324 | 49324 | 49324 | 49324 | 49324 | 49324 | 49324 | 49324 | 49324 | 49324 |
| piperout-08 | 125055 | 125055 | 125055 | 133707 | 125055 | 125055 | 125055 | 125055 | 125055 | 125055 |
| piperout-27 | 8124 | 8124 | 8124 | 8124 | 8124 | 8124 | 8124 | 8124 | 8124 | 8124 |
| pk1 | 11 | 11 | 11 | 12 | 11 | 11 | 11 | 11 | 11 | 11 |
| proteindesign121hz512p9 | 2609 | 2609 | 2609 | 2609 | 2609 | 2609 | 2609 | 2609 | 1477 | 1473 |
| proteindesign122trx11p8 | 2916 | 2916 | 2916 | 2916 | 2916 | 2916 | 1748 | 1762 | 1748 | 1747 |
| qap10 | 340 | 340 | 340 | 340 | 340 | 340 | 340 | 340 | 340 | 340 |
| radiationm18-12-05 | 19527 | 18874 | 18874 | 19853 | 19527 | 19202 | 17566 | 17569 | 17567 | 17566 |
| radiationm40-10-02 | 235396 | 155354 | 155354 | 235396 | 235396 | 209796 | 155330 | 156939 | 155331 | 155328 |
| rail01 | -69.09 | -69.09 | -69.09 | -69.89 | -69.89 | -69.09 | -69.89 | -69.89 | -70.57 | -70.57 |
| rail507 | 174 | 174 | 174 | 178 | 175 | 174 | 174 | 174 | 174 | 174 |
| ran14x18-disj-8 | 3715 | 3714 | 3712 | 3798 | 3798 | 3715 | 3712 | 3712 | 3736 | 3712 |
| rd-rplusc-21 | 179751.8 | 179751.8 | 179751.8 | 179836.5 | 179751.8 | 179751.8 | 165395.3 | 165395.3 | 165395.3 | 165395.3 |
| reblock115 | -36721080 | -36799530 | -36800600 | -36777270 | -36799530 | -36799530 | -36800603 | -36800603 | -36800603 | -36800603 |
| rmatr100-p10 | 423 | 423 | 423 | 442 | 424 | 457 | 423 | 423 | 423 | 423 |
| rmatr200-p5 | 5489 | 5489 | 4521 | 5489 | 5489 | 5489 | 4521 | 4521 | 4521 | 4521 |
| rocI-4-11 | -6020203 | -6020203 | -6020203 | -5040303 | -5040303 | -6020203 | -6020203 | -6020203 | -6020203 | -6020203 |
| rocII-5-11 | -4.65 | -5.66 | -5.67 | -4.65 | -5.66 | -4.65 | -6.68 | -6.68 | -5.68 | -6.68 |
| rococoB10-011000 | 19988 | 19879 | 19534 | 19701 | 19879 | 19988 | 19449 | 19449 | 19497 | 19449 |
| rococoC10-001000 | 11530 | 11460 | 11460 | 11576 | 11472 | 11460 | 11460 | 11460 | 11460 | 11460 |
| roi2alpha3n4 | -61.37 | -63.17 | -63.17 | -62.41 | -62.41 | -63.17 | -63.21 | -63.21 | -63.21 | -63.21 |

| Instance | SCIP | SCIP(600s) | SCIP(900s) | U-LNS | R-LNS | FT-LNS | BTBS-LNS | BTBS-LNS-F | Gurobi | Optimal Solution |
|---|---|---|---|---|---|---|---|---|---|---|
| roi5alpha10n8 | -44.89 | -45.15 | -45.15 | -44.36 | -44.89 | -44.89 | -52.28 | -52.28 | -50.59 | -52.3222744 |
| roll3000 | 12890 | 12890 | 12890 | 12902 | 12890 | 12890 | 12890 | 12890 | 12890 | 12890 |
| s100 | 0 | 0 | 0 | 0 | 0 | 0 | -0.16966 | -0.16966 | -0.03945 | -0.169723527 |
| s250r10 | -0.1437 | -0.1698 | -0.1708 | -0.1437 | -0.1437 | -0.1437 | -0.17178 | -0.17178 | -0.17178 | -0.17178 |
| satellites2-40 | 49 | 49 | 49 | 49 | 49 | 49 | -19 | -19 | -19 | -19 |
| satellites2-60-fs | 28 | 28 | 27 | 27 | 27 | 27 | -19 | -19 | -19 | -19 |
| savsched1 | 31846.3 | 31846.3 | 31846.3 | 45875.9 | 45875.9 | 31846.3 | 3265 | 3265 | 3218 | 3218 |
| sct2 | -230.91 | -230.99 | -230.99 | -230.78 | -230.85 | -230.91 | -230.99 | -230.99 | -230.99 | -230.99 |
| seymour | 427 | 425 | 423 | 427 | 428 | 427 | 423 | 423 | 423 | 423 |
| seymour1 | 410.76 | 410.76 | 410.76 | 410.76 | 410.76 | 410.76 | 410.76 | 410.76 | 410.76 | 410.76 |
| sing326 | 7833336 | 7765711 | 7765711 | 7833336 | 7833336 | 7833336 | 7753675 | 7753675 | 7753676 | 7753675 |
| sing44 | 8175655 | 8174767 | 8174767 | 8177833 | 8163698 | 8175655 | 8128831 | 8128831 | 8130643 | 8128831 |
| snp-02-004-104 | 586912700 | 586816300 | 586804500 | 586829700 | 587089300 | 586821500 | 586803239 | 586803239 | 586803239 | 586803239 |
| sorrell3 | -11 | -15 | -15 | -11 | -15 | -15 | -16 | -16 | -16 | -16 |
| sp150x300d | 69 | 69 | 69 | 69 | 69 | 70 | 69 | 69 | 69 | 69 |
| sp97ar | 688832800 | 682989900 | 681332100 | 681332100 | 673491900 | 679524100 | 660705646 | 660834000 | 660705646 | 660705646 |
| sp98ar | 537245600 | 533010800 | 532905600 | 533010800 | 533455300 | 532891300 | 529740623 | 529905800 | 529740623 | 529740623 |
| splice1k1 | -73 | -121 | -394 | -121 | -121 | -121 | -394 | -394 | -338 | -394 |
| square41 | 26 | 26 | 26 | 26 | 21 | 21 | 15 | 17 | 16 | 15 |
| square47 | 29 | 29 | 29 | 21 | 21 | 21 | 18 | 20 | 20 | 16 |
| supportcase10 | 19 | 19 | 19 | 9 | 19 | 19 | 8 | 8 | 8 | 7 |
| supportcase12 | -7430.15 | -7437.1 | -7475.67 | -7351.97 | -7436.17 | -7449.13 | -7543.26 | -7543.26 | -7559.2419 | -7559.2419 |
| supportcase18 | 49 | 49 | 49 | 50 | 51 | 49 | 48 | 48 | 49 | 48 |
| supportcase26 | 1781.003 | 1747.033 | 1747.033 | 1755.845 | 1768.264 | 1768.264 | 1755.525 | 1755.525 | 1745.124 | 1745.124 |
| supportcase33 | -345 | -345 | -345 | -340 | -345 | -345 | -345 | -345 | -345 | -345 |
| supportcase40 | 24478.86 | 24465.78 | 24465.78 | 24465.78 | 24478.86 | 24294.09 | 24256.31 | 24256.31 | 24256.31 | 24256.31 |
| supportcase42 | 8.0904 | 8.0019 | 7.7683 | 7.7678 | 7.7685 | 7.7811 | 7.7586 | 7.7713 | 7.7586 | 7.7586 |
| supportcase6 | 51921.76 | 51921.76 | 51921.76 | 51921.76 | 51921.76 | 51906.48 | 51906.48 | 51906.48 | 51906.48 | 51906.48 |
| supportcase7 | -1132.223 | -1132.223 | -1132.223 | -1129.28 | -1132.223 | -1132.223 | -1132.223 | -1132.223 | -1132.223 | -1132.223 |
| swath1 | 379.07 | 379.07 | 379.07 | 379.07 | 381.51 | 379.07 | 379.07 | 379.07 | 379.07 | 379.07 |
| swath3 | 397.76 | 397.76 | 397.76 | 399.33 | 397.76 | 397.76 | 397.76 | 397.76 | 397.76 | 397.76 |
| tbfp-network | 131.88 | 24.16 | 24.16 | 25.12 | 24.91 | 24.16 | 24.16 | 24.16 | 24.16 | 24.16 |
| thor50dday | 59310 | 59310 | 40432 | 40432 | 40432 | 40432 | 40417 | 40417 | 40417 | 40417 |
| timtab1 | 764772 | 764772 | 764772 | 766166 | 766345 | 764772 | 764772 | 764772 | 764772 | 764772 |
| tr12-30 | 130596 | 130596 | 130596 | 130608 | 130596 | 130608 | 130596 | 130596 | 130596 | 130596 |
| traininstance2 | 79180 | 77420 | 77420 | 79180 | 84090 | 79180 | 71820 | 72950 | 71820 | 71820 |
| traininstance6 | 29420 | 28460 | 28460 | 28290 | 28460 | 29250 | 28290 | 29250 | 28290 | 28290 |
| trento1 | 25255630 | 18223810 | 18223810 | 18223810 | 7282245 | 15981790 | 5189487 | 5191562 | 5189487 | 5189487 |
| triptim1 | 25.5 | 25.5 | 25.5 | 22.87 | 25.5 | 22.87 | 22.87 | 22.87 | 22.87 | 22.87 |
| uccase12 | 11507.41 | 11507.41 | 11507.41 | 11507.42 | 11507.48 | 11507.41 | 11507.41 | 11507.41 | 11507.41 | 11507.41 |
| uccase9 | 463233.3 | 48328.09 | 15347.75 | 48328.09 | 48328.09 | 20176.81 | 11052.31 | 11052.31 | 10994.13 | 10993.1314 |
| uct-subprob | 315 | 314 | 314 | 317 | 315 | 314 | 314 | 314 | 314 | 314 |
| unitcal_7 | 19635620 | 19635558 | 19635558 | 19635558 | 19635558 | 19635558 | 19635558 | 19635558 | 19635558 | 19635558 |
| var-smallemery-m6j6 | -149.375 | -149.375 | -149.375 | -147.031 | -146.312 | -149.375 | -149.375 | -149.375 | -149.375 | -149.375 |
| wachplan | -8 | -8 | -8 | -8 | -8 | -8 | -8 | -8 | -8 | -8 |

