# OpenReview forum: "BTBS-LNS: Binarized-Tightening, Branch and Search on Learning LNS Policies for MIP"
_ICLR.cc/2025/Conference — ICLR 2025 Poster_

### Official Review · Reviewer_Xeij · 2024-11-04

**Soundness:** 2
**Presentation:** 2
**Contribution:** 2
**Rating:** 5
**Confidence:** 4

**Summary:**

The paper presents an ML-guided LNS framework for MIPs. It combines a binarize-and-tighten scheme to address general integer variables and a branching policy to help escape local optima. It also has a search policy to guide destroy variables. In the experiment, the new method is compared against a variety of ML-guided approaches and heuristic approaches. The presented results show that the proposed method finds better solutions at a faster speed.

**Strengths:**

1. The paper proposes a combination of ML-guided search policy and branching policy for LNS in MIP solving. Various techniques and engineering designs are proposed. The novelty of the paper comes from those details.
2. The effectiveness is demonstrated in experiments with different settings and ablation studies.

**Weaknesses:**

1. The proposed methods are not novel. The global branching idea is similar to [2] by learning from the global (near-)optimal solutions. While the local branching variant is similar to [1]. A detailed explanation of the differences from them should be included.
2. The paper is a little bit hard to follow. It includes a lot of information, it would be good to include a summary of takeaways for each experiment.
3. How are the policies trained in the MIPLIB 2017 experiments? How do you perform training, validation and testing? The variance of the hardness for MIPLIB is large, how do you deal with it when splitting the training, validation and testing set? Can you also summarize the results on MIPLIB in comparison with the other baselines besides Gurobi? Can you bold the best entry in the per-instance result tables in Appendix?


[1] Vinod Nair, et al. Solving mixed integer programs using neural networks. arXiv preprint, 2020

[2] Qingyu Han et al. A gnn-guided predict-and-search framework for mixed-integer linear programming. ICLR, 2023.

**Questions:**

1. What is eq. 6 in figure 2?

---

> ### Author Response · Authors · 2024-11-19
> **Response to reviewer Xeij (1/2)**
>
> Thanks for the comments and valuable suggesstions. We address each point and clarify in our revision at the updated paper version.
>
> > **Q1**: The proposed methods are not novel. The global branching idea is similar to [2] by learning from the global (near-)optimal solutions. While the local branching variant is similar to [1]. A detailed explanation of the differences from them should be included.
>
>
> Thank you for your comments. The main contributions are described in Section 1, including **Bound Tightening for MIP**, **step-by-step incorporation of global information with Global(Local) branching**, and the **problem encoding with an improved attention architecture**. Regarding your primary concern about the comparison between our global (local) branching mechanisms and those in references [1] and [2], we would like to clarify the following points:
>
> 1. **Different Targets and Decision Scopes**: Our branching mechanism serves as a correction module for the RL-based LNS strategy, focusing on whether the LNS policy correctly fixes variables. It uses global-view information to evaluate and refine LNS decisions. While both our approach and [1] and [2] utilize global perspective information, such as optimal solutions or local branching constraints, the scope and target differs. References [1] and [2] make LNS decisions (fix/unfix) for the entire variable set or predict the optimal solution, whereas our branching mechanism complements LNS by correcting its variable fixing results without making decisions for the entire variable set, primarily judging the correctness of the LNS fixing decisions.
> 2. **Different Representations and Architectures**: Our method represents the problem as a tripartite graph, enabling more effective integration of information from the objective function into the decision-making process. By comparison, [1] and [2] employ a bipartite graph structure. Furthermore, we propose an improved attention-based GNN architecture, which demonstrates empirical effectiveness in enhancing the quality of decisions.
>
> > **Q2**: The paper is a little bit hard to follow. It includes a lot of information, it would be good to include a summary of takeaways for each experiment.
>
>
> Thank you for your comments. Below is a detailed summary of the experimental work:
>
> 1. **Datasets**: We conducted tests on both integer programming (IP) and mixed-integer programming (MIP) problems. For IP, we used four well-known NP-hard binary integer programming problems: SC, MIS, CA, and MC. For MIP, we utilized two datasets from the Machine Learning for Combinatorial Optimization (ML4CO) Competition and the MIPLIB2017 benchmark set, a widely recognized standard for MIP solvers.
> 2. **Baselines**: We compared our model against numerous baselines, including commercial and open-source MIP solvers, classical heuristic-based Large Neighborhood Search (LNS) methods, and state-of-the-art learning-based approaches.
> 3. **Evaluation**: Our experiments evaluated the model's capabilities and performance across several dimensions, demonstrating its superior solving ability and stability compared to various baselines:
>     - **Solving efficiency**: Performance on different datasets (Tables 2, 4, 5).
>     - **Scalability**: Ability to handle problems of varying sizes (Table 3).
>     - **Real-world Potential**: Comprehensive evaluation on the full MIPLIB2017 benchmark set, showcasing its applicability in real-world scenarios (Table 6 and Appendix A.8).
>     - **Solver Adaptability**: Compatibility with different baseline solvers, including Gurobi (Appendix A.4) and SCIP.
>     - **Hardware Performance**: Comparison of our BTBS-LNS on CPU and GPU, highlighting potential improvements with GPU optimization (Appendix A.5).
>     - **Anytime Performance**: Model performance at different cut-off time (Appendix A.7).
>     - **Stability**: Consistency and reliability of the model (Appendix A.6).
>
> **(Continue in the next response).**
>
> **References**:
>
> [1] Vinod Nair, et al. Solving mixed integer programs using neural networks. arXiv preprint, 2020
>
> [2] Qingyu Han et al. A gnn-guided predict-and-search framework for mixed-integer linear programming. ICLR, 2023.

---

> > ### Author Response · Authors · 2024-11-19
> > **Response to reviewer Xeij (2/2)**
> >
> > **(Continue from the previous response).**
> >
> > > **Q3**: Some clarifications for the experiments on the MIPLIB2017.
> >
> > >> Q.1 How are the policies trained in the MIPLIB 2017 experiments? How do you perform training, validation and testing?
> >
> > As described in Section 4.5, we perform cross-validation to conduct a comprehensive comparison across all benchmark instances. In each round, we randomly divide the instances into training, validation, and testing sets with proportions of 70%, 15%, and 15%, respectively. The batch size for the training set is set to 3, meaning each batch consists of 3 instances, and the batches are randomly shuffled at the beginning of each epoch. We train for 20 epochs per training instance, with 50 iterations per epoch and a 5-second re-optimization time limit per iteration. Other hyperparameters remain unchanged. Both the RL-based LNS and the branching policies are trained on the training instances, and the policy that performs best on the validation set is selected and directly applied to the testing set. This ensures that each instance can be tested.
> >
> > >> Q.2 Can you also summarize the results on MIPLIB in comparison with the other baselines besides Gurobi?
> >
> > Thank you for your comments. In Table 6, we provide a detailed comparison of the average primal gaps for each method. To further evaluate the performance, we also report the number of wins for each method in the following table, defined as the number of instances where the best feasible solution is found within a fixed cut-off time. If multiple methods find the same best feasible solution for a given instance, they are all recorded as winners. The results show that our proposed method produces the best feasible solution on over 80% of the 240 instances, demonstrating its superior performance. Among the other methods, Gurobi is the closest competitor, 5%-10% inferior to our BTBS-LNS, whereas SCIP and other LNS baselines perform significantly worse in comparison.
> >
> >
> > ||SCIP|SCIP(600s)|SCIP(900s)|U-LNS|R-LNS|FT-LNS|BTBS-LNS|BTBS-LNS-F|Gurobi|
> > |:---:|:---:|:---:|:---:|:---:|:---:|:---:|:---:|:---:|:---:|
> > | **Wins**| 96|109|117|49|51|76|**201**|169|191|
> >
> >
> >
> > >> Q.3 Can you bold the best entry in the per-instance result tables in the Appendix?
> >
> > Thank you for your suggestions. We have bolded the best entry in each row and marked it in blue in the appendix. Additionally, we have calculated the number of wins for each method, as shown in the table above.
> >
> > > **Q4**: What is eq. 6 in figure 2?
> >
> > Sorry for some typos. We have corrected it to Eq.5 in the updated paper version.

---

> > > ### Comment · Reviewer_Xeij · 2024-11-21
> > > **additional feedback**
> > >
> > > Thanks for the rebuttal.
> > >
> > > I have some additional feedback. Upon reviewing the related work, it seems that the method is not compared with some recent state-of-the-arts (as other reviewers have also pointed out):
> > >
> > > 1. https://proceedings.mlr.press/v244/kong24a.html
> > > 2. https://proceedings.mlr.press/v202/ye23e.html (this one seems to have code available)
> > >
> > > Some of the cited work in the paper was not compared with either.

---

> > > > ### Author Response · Authors · 2024-11-22
> > > > **Further Response to reviewer Xeij**
> > > >
> > > > Thank you for your further feedback. Additional experimental results with GNN-GBDT [2] and CL-LNS [3] on the SC and CA datasets are presented in the following table. Implementation details are described in our responses to each specific question. **These results should offer strong evidence of the effectiveness of our method.**
> > > > It is worth noting that the additional experiments have already been completed before the rebuttal. Further experimental supplements on other datasets may take additional time, while we will try to supplement them as soon as possible.
> > > >
> > > > We would like to further clarify that the current experiments provide compelling evidence of our algorithm's effectiveness, as noted by some reviewers. In this paper, we propose a general LNS framework for MIP problems and test it on the MIPLIB2017 benchmark set, which includes real-world problems with various variable types and distributions. Few previous studies have tested on the entire MIPLIB2017 benchmark set, highlighting the practical potential of our method. Additionally, purely taking SCIP as the baseline solver, our BTBS-LNS outperforms Gurobi, a SOTA MIP solver, on numerous datasets, including MIPLIB2017, further revealing the effectiveness. In contrast, [1] uses Gurobi as the baseline solver and evaluates it on only 35 selected MIPLIB2017 instances.
> > > >
> > > > **Set Covering(SC)**
> > > > ||Gap%|Primal Integral (PI)|
> > > > |:---:|:---:|:---:|
> > > > |GNN-GBDT|1.78|18169|
> > > > |CL-LNS|0.92|17025|
> > > > |BTBS-LNS-L|0.47|16234|
> > > > |BTBS-LNS-G|**0.35**|**16205**|
> > > > |Gurobi|0.75|16796|
> > > >
> > > > **Generalization to SC2**
> > > > ||Gap%|PI|
> > > > |:---:|:---:|:---:|
> > > > |GNN-GBDT|1.78|13069|
> > > > |CL-LNS|1.41|12914|
> > > > |BTBS-LNS-L|**0.51**|**12431**|
> > > > |BTBS-LNS-G|0.68|12498|
> > > > |Gurobi|0.71|12528|
> > > >
> > > > **Generalization to SC4**
> > > > ||Gap%|PI|
> > > > |:---:|:---:|:---:|
> > > > |GNN-GBDT|3.45|14169|
> > > > |CL-LNS|3.39|14325|
> > > > |BTBS-LNS-L|**0.84**|**13716**|
> > > > |BTBS-LNS-G|1.20|13789|
> > > > |Gurobi|1.22|13795|
> > > >
> > > > **Combinatorial Auction(CA)**
> > > > ||Gap%|PI(×$10^3$)|
> > > > |:---:|:---:|:---:|
> > > > |GNN-GBDT|2.24|2206.9|
> > > > |CL-LNS|2.05|2198.5|
> > > > |BTBS-LNS-L|2.18|2196.8|
> > > > |BTBS-LNS-G|**1.43**|**1998.9**|
> > > > |Gurobi|1.44|2075.4|
> > > >
> > > > **Generalization to CA2**
> > > > ||Gap%|PI(×$10^3$)|
> > > > |:---:|:---:|:---:|
> > > > |GNN-GBDT|3.44|5508.9|
> > > > |CL-LNS|3.51|5621.7|
> > > > |BTBS-LNS-L|**1.67**|**4800.3**|
> > > > |BTBS-LNS-G|1.89|5012.6|
> > > > |Gurobi|3.60|5723.5|
> > > >
> > > > **Generalization to CA4**
> > > > ||Gap%|PI(×$10^3$)|
> > > > |:---:|:---:|:---:|
> > > > |GNN-GBDT|2.86|12853|
> > > > |CL-LNS|2.99|13025|
> > > > |BTBS-LNS-L|**1.39**|**11128**|
> > > > |BTBS-LNS-G|1.46|11705|
> > > > |Gurobi|12.61|21959|
> > > >
> > > > > **Q1**: It seems that the method is not compared with some recent state-of-the-arts Kong et al.[1] and Ye et al.[2].
> > > >
> > > > Thanks for your comments and valuable insights.
> > > >
> > > > **Regarding Reference [1]**
> > > >
> > > > We have not yet compared with [1] as it is relatively new, and the code has not been open-sourced. However, we have comprehensively evaluated our method on the datasets mentioned in [1], including the full MIPLIB2017 benchmark set. Reference [1] tested only 35 instances with integer variables. Our results demonstrate the superior performance of our method compared to LNS baselines and even Gurobi, despite using SCIP as the baseline solver. We will consider incorporating a further comparison in our final paper version.
> > > >
> > > > **Regarding Reference [2]**
> > > >
> > > > The problem addressed in [2] differs from ours. It focuses on predicting feasible solutions and then iteratively optimizing them using rule-based LNS policies. In contrast, our work concentrates on searching for feasible solutions given an initial solution, using learning-based LNS approaches. However, since the code is available, we have recently evaluated its performance on our datasets. Note that some key algorithmic details, such as graph partitioning and the GCN architecture, are not fully disclosed. Therefore, we had to rely on the details provided in the paper to reproduce. The performance of [2] (denoted as GNN-GBDT) on the SC and CA datasets are summarized in the table above.
> > > >
> > > > > **Q2**: Some of the cited work in the paper was not compared with either.
> > > >
> > > > Thank you for your comments. Most of the cited works related to our research on LNS have already been compared, demonstrating the effectiveness of our proposed method. One notable exception, which has received significant attention in recent literature, is CL-LNS [3], also used for comparison in reference [1]. Other cited works have either been compared in our paper or have been shown to perform worse than CL-LNS [3].
> > > >
> > > > We have evaluated the performance of CL-LNS [3] on the SC and CA datasets using their official open-source code. The results are summarized in the table above.
> > > >
> > > > **References**:
> > > >
> > > > [1] Kong et al. "ILP-FORMER: Solving Integer Linear Programming with Sequence to Multi-Label Learning." UAI 2024.
> > > >
> > > > [2] Ye H et al. GNN&GBDT-guided fast optimizing framework for large-scale integer programming//ICML 2023.
> > > >
> > > > [3] Huang T et al. Searching large neighborhoods for integer linear programs with contrastive learning//ICML 2023.

---

> > > > > ### Author Response · Authors · 2024-11-26
> > > > > **Supplementary Complete Experimental Results**
> > > > >
> > > > > We are deeply grateful for your valuable suggestions. In response to the previous feedback, we have supplemented the complete experimental results for the two additional baselines, GNN-GBDT [1] and CL-LNS [2], in Tables 2 and 3 of the updated paper. These updates are highlighted in blue for easy reference. The results clearly demonstrate that our proposed BTBS-LNS outperforms both GNN-GBDT and CL-LNS across all datasets, further substantiating its effectiveness.
> > > > >
> > > > > Moreover, to provide a more comprehensive evaluation of our method, we are diligently working on reproducing the ILP-former [3], as recommended by you and reviewer aDL4. Given the lack of publicly available code, we are meticulously reconstructing this method based on the detailed descriptions provided in the original paper. We aim to complete this reproduction process as soon as possible and will integrate the full results into the final version of our paper.
> > > > >
> > > > > We sincerely appreciate your insightful guidance. Please do not hesitate to contact us if you have any further questions or require additional information. We hope this will contribute to a more thorough evaluation of our work.
> > > > >
> > > > >
> > > > > **References**:
> > > > >
> > > > > [1] Ye H et al. GNN&GBDT-guided fast optimizing framework for large-scale integer programming//ICML 2023.
> > > > >
> > > > > [2] Huang T et al. Searching large neighborhoods for integer linear programs with contrastive learning//ICML 2023.
> > > > >
> > > > > [3] Kong et al. "ILP-FORMER: Solving Integer Linear Programming with Sequence to Multi-Label Learning." UAI 2024.

---

> > > > > > ### Author Response · Authors · 2024-11-29
> > > > > > **Additional experimental results on ILP-FORMER**
> > > > > >
> > > > > > Following the suggestions from you and reviewer aDL4, we have been working diligently to reproduce ILP-FORMER [1], and the extended rebuttal period has allowed us to provide the latest experimental results promptly. Given that the code for ILP-former is not publicly available, we reproduced it based on the framework and algorithmic details described in the paper.
> > > > > >
> > > > > > To ensure a fair comparison with other methods, we used SCIP as the baseline solver for the data collection and inference, starting the search from the first feasible solution found by SCIP, consistent with the experimental setup in our paper. All other parameter configurations are aligned with the default parameters in [1]. We have completed the training and evaluation on the Set Covering(SC) dataset. The comparison results are as follows:
> > > > > >
> > > > > >
> > > > > > **Set Covering(SC)**
> > > > > > ||Gap%|Primal Integral (PI)|
> > > > > > |:---:|:---:|:---:|
> > > > > > |RL-LNS|1.29|17623|
> > > > > > |CL-LNS|0.92|17025|
> > > > > > |ILP-FORMER|0.85|16893|
> > > > > > |BTBS-LNS-L|0.47|16234|
> > > > > > |BTBS-LNS-G|**0.35**|**16205**|
> > > > > > |Gurobi|0.75|16796|
> > > > > >
> > > > > >
> > > > > > **Generalization to SC2**
> > > > > > ||Gap%|Primal Integral (PI)|
> > > > > > |:---:|:---:|:---:|
> > > > > > |RL-LNS|1.66|13007|
> > > > > > |CL-LNS|1.41|12914|
> > > > > > |ILP-FORMER|1.29|12898|
> > > > > > |BTBS-LNS-L|**0.51**|**12431**|
> > > > > > |BTBS-LNS-G|0.68|12498|
> > > > > > |Gurobi|0.71|12528|
> > > > > >
> > > > > >
> > > > > > **Generalization to SC4**
> > > > > > ||Gap%|Primal Integral (PI)|
> > > > > > |:---:|:---:|:---:|
> > > > > > |RL-LNS|3.73|14866|
> > > > > > |CL-LNS|3.39|14325|
> > > > > > |ILP-FORMER|3.18|14693|
> > > > > > |BTBS-LNS-L|**0.84**|**13716**|
> > > > > > |BTBS-LNS-G|1.20|13789|
> > > > > > |Gurobi|1.22|13795|
> > > > > >
> > > > > >
> > > > > > From the results, ILP-FORMER consistently outperforms RL-LNS and CL-LNS, which aligns with the evaluation results in their original paper. Additionally, our BTBS-LNS consistently outperforms ILP-FORMER on the Set Covering (SC) dataset and its variants with increased constraints and variables (SC2 and SC4), further validating the effectiveness of our approach.
> > > > > >
> > > > > > It is worth noting that due to the time required for data collection and training, which can take several days, we are unable to provide comparison results for the remaining datasets within the rebuttal period. However, we commit to including the full comparison results in the final version of the paper.
> > > > > >
> > > > > > Thank you for your understanding. Please do not hesitate to contact us if you have any further questions or require additional information.
> > > > > >
> > > > > >
> > > > > > **References**:
> > > > > >
> > > > > > [1] Kong et al. "ILP-FORMER: Solving Integer Linear Programming with Sequence to Multi-Label Learning." UAI 2024.

---

### Official Review · Reviewer_bYsh · 2024-11-04

**Soundness:** 3
**Presentation:** 2
**Contribution:** 3
**Rating:** 8
**Confidence:** 4

**Summary:**

Authors propose a neural large neighborhood search (LNS) policy which improves upon previously proposed methods in mainly three aspects. First, authors focus on general mixed integer programming (MIP) which some variables can be integer type (as opposed to be only allowed binary type), and proposes a "binarized tightening" method, which performs much better than previous methods. Second, authors propose to remove softmax operation in the graph neural network for MIPs. Third, authors train additional LNS policies with imitation learning, and let these policies override decisions from RL-trained LNS policy. These imitation-learned policies provide additional signals and hence help escaping from local minimum.

**Strengths:**

Significance: The proposed method demonstrates a strong empirical performance compared against a large number of previous learning-based methods as well as strong traditional solvers (Gurobi, SCIP). Such a strong empirical result can drive a stronger adoption of learning-based MIP solvers in practice. The coverage of experiments and the experimental protocol is high quality (discussed below in the quality section), which will give practitioners high confidence to adopt the proposed approach. If successful, this will make a significant impact to applications of MIP.

Quality: The coverage of benchmarks is very high. Authors not only experiment with synthetic tasks popular in previous papers, but also more recent and challenging benchmarks in ML4CO competition and MIPLIB2017. Authors compare against 13 learning-based baselines, which comprehensively validates algorithmic design decisions made in this paper. Although the proposed method is quite complex, such a strong validation justifies the complexity well.

Originality: Binarized tightening is a novel technique clearly differentiated from previously proposed methods for general MIPs. Other techniques introduced are not very surprisingly new, but still contributes significantly to the performance of the final approach, and validated well by extensive experiments.

Clarity: Although the proposed algorithm is quite complex with many components, authors do a good job clearly explaining them, and providing good justifications both conceptually and empirically for every algorithmic design decision they make. Some terminologies are phrased a bit confusingly, however, which I elaborate in the Weaknesses section.

**Weaknesses:**

Authors argue their branching network leverages "global" information and uses a "global" view, but I believe this is a misnomer because at the inference time, the branching network uses the same state representations as the LNS policy. It is just that the branching network is _trained_ with different imitation learning signals, global optimum solution for BTBS-LNS-G and local branching for BTBS-LNS-L. Other than the training method (RL vs. imitation learning), the branching network makes the same decisions (whether to fix each variable) as the LNS policy do; in fact, what authors call as local branching is called LNS from previous work (Sonnerat et al, 2022). I would consider these as simply additional LNS policies, just trained differently (RL vs. imitation learning).

While my major concern is mostly on the phrasing of "global" information because such information is not available at inference time, I also point out that this perspective - that branching networks are simply LNS policy trained differently - offers more natural ways to combine these branching networks into the algorithm. For example, they could be ensembled together using standard ensemble methods, because they are policies for the same objective which are just trained differently. Or, LNS policy could be initialized by these branching policies for transfer learning. I believe these are more natural ways of combining these two policies, rather than first making predictions with RL policy and then override predictions by imitation-learned policies.

There's another suggestion for clarification. In Line 070, authors say: "We further develop an attention-based tripartite graph (Ding et al., 2020)". Although Ding et al (2020) is attributed here, the statement can be read as attention-based tripartite graph being proposed by authors for the first time. To make the attribution clearer, I would suggest to rephrase as something like: "We employ an attention-based tripartite graph (Ding et al., 2020). We improve the previously proposed attention architecture by removing the softmax normalization, and demonstrate its empirical effectiveness." Such statement will make authors' contribution clearer.

**Questions:**

How are substitute variables for integer variables represented?  $j$ in $a_{i,j}^t$ should be reflected as feature somewhere, but features in Table 7 does not seem to discuss it.

---

> ### Author Response · Authors · 2024-11-19
> **Response to reviewer bYsh**
>
> We sincerely thank Reviewer bYsh for the positive feedback and precious suggestions! We have addressed each point and provided clarifications in the updated version of the paper.
>
>
> > **Q1**: Clarify the "global" information in the branching.
>
> Thank you for your insightful comments. We appreciate the opportunity to clarify the meaning of "global information" in our work.
>
> The term "global information" refers to the broader context that goes beyond the immediate observations available during the RL-based training and inference processes. In our framework, the network can observe the current and historical feasible solutions and leverage the problem structure (tripartite graph) to make decisions about fixing/unfixing variables. However, this process may lack a comprehensive evaluation of the current solving state, such as the distance to the optimal solution or the proximity of certain variables to their optimal values. These "global" perspectives provide additional insights that can enhance the accuracy of LNS decisions. Therefore, we introduced an additional branching network to incorporate these global perspectives, aiming to refine the LNS decisions from a holistic problem-solving viewpoint.
>
> We fully acknowledge your point that these "global" perspectives are not directly observable during the inference stage. Consequently, we use offline training to capture the relationship between local decisions and global perspectives. Specifically, the training signals indicate whether the LNS policy produces a correct fixing compared to the optimal solution. The trained network is then applied during inference to make more informed decisions. It differs slightly from the LNS policy in that it focuses on the LNS fixed variable set, rather the entire variable space. Essentially, the branching network functions as a strategy correction mechanism for LNS, leveraging global perspectives during training and applying them in inference.
>
>
> > **Q2**: Multiple policies can be better ensembled together using standard ensemble methods, because they are policies for the same objective which are just trained differently. Or, LNS policy could be initialized by these branching policies for transfer learning.
>
>
> Thank you for your insightful comments. We appreciate your suggestions regarding the integration of branching networks into LNS.
>
> In our current implementation, the branching network serves as a strategy correction module for the LNS policy. It is an empirical solution that has proven effective in practical applications. However, we fully recognize the value of your suggestions for more natural ways to combine these policies. Your ideas, such as using standard ensemble methods or initializing the LNS policy with the branching policies for transfer learning, offer promising avenues for more efficiently integrating different strategies. These approaches could potentially enhance the robustness and performance of our framework, especially in real-world scenarios. We agree that these are more natural ways and plan to explore these methods in our future research.
>
>
> > **Q3**: Clarification for the contributions in the attention-based tripartite graph.
>
> Thank you for your valuable suggestions. We have updated the descriptions in the revised version of the paper, with the changes marked in blue.
>
> > **Q4**: How are substitute variables for integer variables represented? Features in Table 7 does not seem to discuss it.
>
> We apologize for omitting some key information. The encoded substitute variables are also characterized by static and dynamic features. For static features, the variable type and bounds are set to binary and 1/0, respectively, while other features are directly inherited from the original integer variables. For dynamic features, the solution value for each substitute variable is determined based on the encoded results. For example, for a general integer variable with a range of [0, 7], if the current solution value is 5, the solution values of the three substitute variables would be 1, 0, 1. Additionally, the connections between the substitute variables and other nodes are directly inherited from the original integer variables.
>
> We have included these details in Appendix A.2 of the revised version of the paper, with the changes marked in blue.

---

> > ### Comment · Reviewer_bYsh · 2024-11-19
> >
> > Thank you for the clarifications! Regarding Q1, yes I understand the intention. I am just worried about this naming "global information" will cause confusions, but I will leave the decision up to authors. This will not impact the score I will vote.
> >
> > And thanks for clarifications on Q2/Q3/Q4. I hope adding the clarification has improved the paper. I will keep my score as I was expecting my questions to be well-addressed when writing the original review.

---

> > > ### Author Response · Authors · 2024-11-20
> > > **Thanks for your recognition**
> > >
> > > Thank you for your valuable feedback.
> > >
> > > Regarding Q1, we understand your concern about the term "global information" and will carefully consider revising it to avoid any potential confusion, or we can add detailed notes to clarify and prevent any misunderstandings. We appreciate your suggestion.
> > >
> > > Thank you again for your positive response and constructive comments.

---

### Official Review · Reviewer_aDL4 · 2024-11-08

**Soundness:** 3
**Presentation:** 3
**Contribution:** 2
**Rating:** 6
**Confidence:** 4

**Summary:**

The paper presents the BTBS-LNS (Binarized-Tightening, Branch and Search for Large Neighborhood Search), a method designed to solve large-scale Mixed Integer Programming (MIP) problems. It addresses two main issues in learning-based LNS approaches: limited exploration in search spaces, and a tendency to prematurely fix important variables, which may lead to suboptimal solutions. The proposed method introduces three main innovations:

Binarized Tightening: Encodes integer variables into binary form, allowing bound tightening to reduce solution space complexity.
Attention-based Tripartite Graph: Models relationships among variables, constraints, and objectives to capture global correlations.
Branching Network: Optimizes potentially misclassified “backdoor” variables to help escape local optima.
Through experiments on benchmarks like MIPLIB2017, the method demonstrates competitive performance, often surpassing traditional solvers like SCIP and sometimes outperforming Gurobi.

**Strengths:**

Innovation in Problem Encoding: Using a tripartite graph structure to represent MIP instances is a novel approach that strengthens the model's ability to capture complex dependencies.

Generalization to Integer Variables: Unlike many LNS methods that focus only on binary variables, BTBS-LNS can handle a broader class of integer variables, improving the method's applicability to general MIP problems.

Superior Performance: The method shows strong empirical results across multiple MIP problems and benchmarks, achieving lower primal gaps than SCIP and Gurobi in some cases.

Adaptive Bound-Tightening: Binarized tightening provides an efficient exploration-exploitation balance, with flexibility for bounded and unbounded variables.

**Weaknesses:**

High Complexity: The approach combines multiple advanced techniques, potentially making it computationally intensive, particularly in scenarios requiring extensive parameter tuning or high-dimensional data.

Dependence on Initialization: The method relies on an initial feasible solution generated by baseline solvers. Poor initialization could impact overall performance, as the effectiveness of the LNS-based policy might depend on the quality of the starting solution.

Local Optima: While the branching network mitigates local optima issues, the reliance on imitation learning and reinforcement learning can still lead to convergence to suboptimal solutions in complex or highly irregular MIP landscapes.

**Questions:**

How does the performance of BTBS-LNS scale with increasing problem size, especially compared to state-of-the-art solvers like Gurobi, in terms of both time complexity and solution accuracy?

Does the choice of baseline solver (SCIP or Gurobi) for generating initial solutions significantly influence BTBS-LNS's overall performance?

How does the model handle diverse real-world MIP problems where variables might not align as neatly with the tripartite structure employed in BTBS-LNS?

Have you compared your work with a recent related work by Kong et al.? https://proceedings.mlr.press/v244/kong24a.html

---

> ### Author Response · Authors · 2024-11-19
> **Response to reviewer aDL4 (1/2)**
>
> Thank you for your valuable comments. We have addressed each point and provided clarifications in our revision.
>
> > **Q1**: High Complexity: The approach combines multiple advanced techniques, potentially making it computationally intensive, particularly in scenarios requiring extensive parameter tuning or high-dimensional data.
>
>
> Thank you for your insights. Our framework integrates RL-based LNS with imitation learning-based branching strategies. To address the complexities in training and inference, we implemented the following optimizations:
>
> 1. **Training on Small-Scale Instances**: We trained the model on relatively simple instances and then applied the learned policies to more complex problems. This approach significantly reduces training complexity and enhances applicability to high-dimensional real-world data.
> 2. **Time-Limited Data Collection**: For label collection in the offline branching training, we did not require globally optimal solutions. Instead, we limited the solving time to 200 seconds to obtain high-quality feasible solutions for training.
> 3. **Generalized Hyperparameter Settings**: We avoided dataset-specific hyperparameter tuning. Most parameters remained consistent across different datasets, as detailed in Section 4, to enhance the generalizability and usability of our framework.
> 4. **Time-Constrained Testing**: All experiments were conducted within a specified time limit, taking into account the inference time of all learned policies. The experimental results demonstrate the time efficiency of our approach.
>
>
> > **Q2**: Dependence on Initialization: The method relies on an initial feasible solution generated by baseline solvers. Poor initialization could impact overall performance, as the effectiveness of the LNS-based policy might depend on the quality of the starting solution.
>
>
> Thank you for your insightful comments. The quality of the initial feasible solution can indeed influence the performance of the performance. However, starting the search from an initial feasible solution is a common practice in LNS, allowing for the exploration and optimization of nearby solutions. In our implementation, we use the first feasible solution found by SCIP as the initial point. To ensure a fair comparison, all LNS methods, except for the MIP solvers (SCIP and Gurobi), start from the same initial point. As demonstrated in Tables 2, 3, 4, 5, and 6, BTBS-LNS consistently exhibits superior performance, even outperforming Gurobi on some instances, including the MIPLIB2017 benchmark set.
>
> To further assess the impact of different initial solutions, we conducted additional experiments in Appendix A.4, where we replaced the baseline solver from SCIP to Gurobi, which may theoretically provide a better initial feasible solution for each instance. The results indicate that a better initial point can help improve the final solution quality to some extent, while our BTBS-LNS has consistently demonstrated its effectiveness over all the competing baselines.
>
>
> > **Q3**: Local Optima: While the branching network mitigates local optima issues, the reliance on imitation learning and reinforcement learning can still lead to convergence to suboptimal solutions in complex or highly irregular MIP landscapes.
>
>
> Thank you for your valuable insights. Our proposed method, which incorporates branching on top of LNS, helps the learned policies to escape local optima to some extent. Figure 4 visually illustrates this point, showing that branching can refine the decisions made by LNS, particularly in the later stages of optimization. Extensive experimental results also demonstrate the effectiveness of our framework.
>
> However, as you pointed out, due to the nature of local search, even the combination of LNS and branching may not completely avoid issues related to local optima. Addressing this challenge will be a key direction for our future research.
>
>
> > **Q4**: How does the performance of BTBS-LNS scale with increasing problem size, especially compared to state-of-the-art solvers like Gurobi, in terms of both time complexity and solution accuracy?
>
>
> In Table 3, we detail the transferability of our learned policies across different problem scales. Specifically, we apply policies learned from smaller instances to larger ones with 2 to 4 times the number of variables and constraints, while maintaining the same distribution (see Table 8 for details). The cutoff solving time is set to 200 seconds, and the inference time for both policies are taken into account. The results show that our proposed method consistently outperforms all LNS baselines and the open-source solver SCIP across different datasets. Compared to Gurobi, our method yields better results on SC2, SC4, CA2, CA4, and MC4, and is only slightly inferior on the remaining three groups.
>
> **(Continue in the next response)**

---

> > ### Author Response · Authors · 2024-11-19
> > **Response to reviewer aDL4 (2/2)**
> >
> > **(Continue from the previous response).**
> >
> > > **Q5**: Does the choice of baseline solver (SCIP or Gurobi) for generating initial solutions significantly influence BTBS-LNS's overall performance?
> >
> > In Appendix A.4, we provide a detailed analysis of the performance when using Gurobi as the baseline solver. Compared to the SCIP counterparts presented in the main text, the results show that, on the same problems, a better initial solution (Gurobi vs. SCIP) often leads to improved performance. For example, in the Set Covering problem, BTBS-LNS, when using Gurobi as the baseline solver, finds better feasible solutions in over 70% of the instances within the same cut-off time, resulting in an overall improvement in solution quality of 2%, while our BTBS-LNS has consistently demonstrated its effectiveness over all the competing baselines.
> >
> >
> > > **Q6**: How does the model handle diverse real-world MIP problems where variables might not align as neatly with the tripartite structure employed in BTBS-LNS?
> >
> >
> > Thank you for your valuable insights. Our proposed framework is designed to deal with general MIP problems, as MIP problems can typically be well-represented by a tripartite graph structure (i.e., variables, constraints, and the objective function), from which relevant features can be extracted. To evaluate the performance of our framework on real-world MIP problems, we conducted experiments on the MIPLIB 2017 benchmark set, which includes instances from various problems, distributions, and scales. This benchmark, generated from various real-world problems, is a well-established standard for evaluating MIP solvers. As shown in Table 6 and Appendix A.8, our proposed method consistently outperforms all competing baselines, including Gurobi, demonstrating its potential for application to real-world MIP problems.
> >
> >
> > > **Q7**: Have you compared your work with a recent related work by Kong et al.[1].
> >
> > Thank you for your suggestions. We have not yet compared our work with the referenced article because it is relatively new, and the code has not been open-sourced, making it challenging to reproduce and compare results. Additionally, the experiments in [1] focused on four generated integer programming problems and 35 instances from MIPLIB2017 that contain only integer variables. In contrast, our BTBS-LNS has been evaluated on these datasets and even on the full MIPLIB2017 benchmark set, which includes 240 instances. The results show that our BTBS-LNS performs competitively with, and sometimes better than Gurobi, even when using SCIP as the baseline solver, particularly on the full MIPLIB 2017 benchmark.
> >
> > We appreciate your suggestions and will consider incorporating a comparison with [1] in the final version of our paper.
> >
> > **References**:
> >
> > [1] Kong, Shufeng, Caihua Liu, and Carla P. Gomes. "ILP-FORMER: Solving Integer Linear Programming with Sequence to Multi-Label Learning." The 40th Conference on Uncertainty in Artificial Intelligence.

---

> > > ### Author Response · Authors · 2024-11-29
> > > **Additional experimental results on ILP-FORMER**
> > >
> > > Following the suggestions from you and reviewer Xeij, we have been working diligently to reproduce ILP-FORMER [1], and the extended rebuttal period has allowed us to provide the latest experimental results promptly. Given that the code for ILP-former is not publicly available, we reproduced it based on the framework and algorithmic details described in the paper.
> > >
> > > To ensure a fair comparison with other methods, we used SCIP as the baseline solver for the data collection and inference, starting the search from the first feasible solution found by SCIP, consistent with the experimental setup in our paper. All other parameter configurations are aligned with the default parameters in [1]. We have completed the training and evaluation on the Set Covering(SC) dataset. The comparison results are as follows:
> > >
> > >
> > > **Set Covering(SC)**
> > > ||Gap%|Primal Integral (PI)|
> > > |:---:|:---:|:---:|
> > > |RL-LNS|1.29|17623|
> > > |CL-LNS|0.92|17025|
> > > |ILP-FORMER|0.85|16893|
> > > |BTBS-LNS-L|0.47|16234|
> > > |BTBS-LNS-G|**0.35**|**16205**|
> > > |Gurobi|0.75|16796|
> > >
> > >
> > > **Generalization to SC2**
> > > ||Gap%|Primal Integral (PI)|
> > > |:---:|:---:|:---:|
> > > |RL-LNS|1.66|13007|
> > > |CL-LNS|1.41|12914|
> > > |ILP-FORMER|1.29|12898|
> > > |BTBS-LNS-L|**0.51**|**12431**|
> > > |BTBS-LNS-G|0.68|12498|
> > > |Gurobi|0.71|12528|
> > >
> > >
> > > **Generalization to SC4**
> > > ||Gap%|Primal Integral (PI)|
> > > |:---:|:---:|:---:|
> > > |RL-LNS|3.73|14866|
> > > |CL-LNS|3.39|14325|
> > > |ILP-FORMER|3.18|14693|
> > > |BTBS-LNS-L|**0.84**|**13716**|
> > > |BTBS-LNS-G|1.20|13789|
> > > |Gurobi|1.22|13795|
> > >
> > >
> > > From the results, ILP-FORMER consistently outperforms RL-LNS and CL-LNS, which aligns with the evaluation results in their original paper. Additionally, our BTBS-LNS consistently outperforms ILP-FORMER on the setcover dataset and its variants with increased constraints and variables (SC2 and SC4), further validating the effectiveness of our approach.
> > >
> > > It is worth noting that due to the time required for data collection and training, which can take several days, we are unable to provide comparison results for the remaining datasets within the rebuttal period. However, we commit to including the full comparison results in the final version of the paper.
> > >
> > > We sincerely appreciate your insightful guidance. Please do not hesitate to contact us if you have any further questions or require additional information.
> > >
> > >
> > > **References**:
> > >
> > > [1] Kong et al. "ILP-FORMER: Solving Integer Linear Programming with Sequence to Multi-Label Learning." UAI 2024.

---

### Official Review · Reviewer_mah2 · 2024-11-12

**Soundness:** 3
**Presentation:** 3
**Contribution:** 3
**Rating:** 6
**Confidence:** 5

**Summary:**

This paper proposes enhancements to an existing ML approach to Large Neighborhood Search (LNS) for Mixed-Integer Linear Programming (MILP). The framework is as follows: value-based RL is used in conjunction with a GNN representation of the MILP to learn a policy that selects, at each step, a subset of the decision variables to fix and search around. The RL reward function favors finding a sequence of improving solutions. This paper aims to improve this framework.

First, the authors devise a heuristic for efficiently handling non-binary integer variables, a common feature of many MILP problems that has not been dealt with in prior ML work. Second, an existing GNN architecture for MILP is slightly modified to allow for better attention scores. Third, a supervised learning approach is used to override potentially incorrect LNS variable fixing actions made by the RL agent. This is done by training another GNN to predict high-quality solutions. Should an LNS fixing action for a variable disagree with the predicted value for it, the LNS action is dismissed.

Experiments are performed on four problem classes as well as some MIPLIB2017 instances. The proposed method along with some ablations of it are compared to solvers SCIP and Gurobi, non-ML LNS methods, and existing ML LNS methods. Favorable performance is exhibited.

**Strengths:**

S1. A significant engineering effort is made to improve the RL-LNS approach. It covers the GNN architecture, the handling of integer variables, and the corrective mechanism. That all these aspects are considered simultaneously is a strength of this paper.

S2. Integer variable handling: Although I have questions about this part, it is a solid contribution to this area of research. Typically, we assume that bounded integers can be recast as a set of binary variables. However, this expansion introduces additional variables and constraints which can slow down MIP solving.

S3. Corrective mechanism: I also liked this contribution given that it addresses limitations of RL.

S4. The experimental setup is sound and the performance of the proposed method is very competitive. It is impressive that it can compete with SCIP and Gurobi.

**Weaknesses:**

W1. The writing can be improved. I have included a sample of minor comments in “Questions” but I do believe that going over the paper using tools such as Grammarly can help improve the clarity of the paper. This is crucial for publication at a conference at the level of ICLR.

W2. The GNN architecture is essentially the same as that of Ding et al.’s. I am not sure that it stands on its own as a contribution. Rather, I view this as an architecture tuning step that is rather incremental.

W3. The corrective mechanism in section 3.4 requires labeled data, namely near-optimal solutions either globally or locally. This increases the data collection cost which now includes this labeling via a MILP solver in addition to RL simulations. Given that the MILPs of interest are NP-Hard to optimize, it is unclear to me that relying on this solving step during data collection is justified. This is a chicken-and-egg situation because if collecting such solutions was easy, no ML would be required. If it is hard, then ML is justified but then this data collection process is potentially prohibitive.

W4. The bound tightening method is not described sufficiently clearly. Perhaps a small numerical example can help Illustrate what you’re doing in Algorithm 1.

**Questions:**

Please refer to the weaknesses above.

The writing can be improved substantially. Here are some minor comments:
- line 148: consider rephrasing to “selecting the variable subsets that may need to be optimized at each step, with the remaining variables fixed or bound-tightened”
- 156: “training details are described in Sec. 3.3. ….”
-  307: “trained offline”
-  Table 2 caption is incomplete?

---

> ### Author Response · Authors · 2024-11-19
> **Response to reviewer mah2**
>
> We appreciate your insightful comments and valuable suggestions. In response, we have addressed each point and provided clarifications in the revised version of the paper.
>
> > **Q1**: The writing can be improved.
>
> Thank you for your valuable suggestions. We have corrected the typos and reviewed the paper using Grammarly to make further improvements, as reflected in the updated paper.
>
> > **Q2**: The GNN architecture is essentially the same as that of Ding et al.’s. I am not sure that it stands on its own as a contribution. Rather, I view this as an architecture tuning step that is rather incremental.
>
> We thank you for your valuable comments. Although the GNN architecture does exhibit similarities with prior works, we have enhanced the attention mechanism by eliminating the softmax normalization, which has been empirically validated to improve performance. To better emphasize our contributions, we have revised the introduction as recommended by the reviewer bYsh, with the modifications highlighted in blue.
>
> > **Q3**: The corrective mechanism in section 3.4 requires labeled data, namely near-optimal solutions either globally or locally. This increases the data collection cost which now includes this labeling via a MILP solver in addition to RL simulations. Given that the MILPs of interest are NP-Hard to optimize, it is unclear to me that relying on this solving step during data collection is justified.
>
> Thank you for your comments. As you pointed out, label collection for NP-Hard MILPs is particularly time-consuming, especially for large-scale instances. To address this, we adopted the following strategies:
>
> 1. **Only in the training:** We collected (near-) optimal solution labels only for the training phase, which typically involves smaller and easier-to-solve instances. The trained policy can then be directly applied to other instances, including large-scale and difficult ones.
> 2. **Solution Quality**: We do not require globally optimal solutions. Instead, we run MIP solvers (e.g., Gurobi) with a restricted cutoff time of 200 seconds. The best incumbent solutions obtained within this time are used as the solution labels. To assess the impact of solution quality, we increased the solving time limit from 200s to 300s on Maximum Cut instances, and potentially better solutions can be obtained to train our method, and other hyperparameters remain unchanged. The results are summarized below.
>
> |                       |     Obj     |   Gap%   |   PI    |
> |:---------------------:|:-----------:|:--------:|:-------:|
> | **BTBS-LNS-G (200s)** |   -922.18   |   0.59   |  785   |
> | **BTBS-LNS-G (300s)** | **-923.05** | **0.42** | **514** |
> |      **Gurobi**       |   -921.90   |   0.62   |  842   |
>
> As evident from the results, BTBS-LNS-G trained with higher-quality solutions achieves slightly better performance.
>
>
>
> > **Q4**: The bound tightening method is not described sufficiently clearly. Perhaps a small numerical example can help Illustrate what you’re doing in Algorithm 1.
>
> Thank you for your suggestions. Here we provide a small numerical example to illustrate the process, and we have included it in Appendix A.9 for clear understanding of Algorithm 1. Consider a general integer variable $x_0$ with a range of $[0, 15]$ and a current solution of $3$. This variable is encoded using 4 substitution binary variables: $x_{0,1}$, $x_{0,2}$, $x_{0,3}$, and $x_{0,4}$.
>
> At each LNS iteration $t$, distinct actions $a_{0,1}^t$, $a_{0,2}^t$, $a_{0,3}^t$, and $a_{0,4}^t$ are taken for each substitution variable, controlling the range of the original variable at different levels of significance. Specifically, the decision actions $a_{0,j}^t$ for all $j$ are evaluated sequentially, and the upper and lower bounds are tightened around the current solution whenever $a_{0,j}^t = 0$, as described in Lines 11-12 of Algorithm 1. Here are some examples:
>
> 1. If $a_{0,1}^t = 0$ and the others are 1, the updated bounds for $x_0$ will be $[0, 10]$.
> 2. If $a_{0,1}^t = a_{0,2}^t = 0$ and the others are 1, the updated bounds for $x_0$ will be $[0, 8]$.
> 3. ...
> 4. If all decision actions are 0, the updated bounds for $x_0$ will be $[0, 6]$, with the current solution precisely at the midpoint of the variable bounds. Similarly, when the current solution is $12$, the updated bounds will be $[9,15]$.
>
> Importantly, this bound-tightening process is conducted independently at each step, always initiating from the original variable bounds.

---

> > ### Comment · Reviewer_mah2 · 2024-11-19
> > **Reading your response**
> >
> > Thank you for responding quickly! I will be reading your answers carefully and follow up as needed.

---

> > ### Comment · Reviewer_mah2 · 2024-11-19
> > **Updated rating**
> >
> > Following your response, I have increased the rating from 5 to 6. If this paper is accepted, please make sure that your publicly available code is easy to run and to reproduce the numerical results reported in the paper. Because you have many datasets, baselines, and ablations, reproducibility will be crucial for the community to be able to build on your work.

---

> > > ### Author Response · Authors · 2024-11-20
> > > **Thanks for your recognition**
> > >
> > > Thank you for increasing the rating and for your valuable feedback. We appreciate your emphasis on reproducibility.  If the paper is accepted, we will ensure that the publicly available code is easy to run and capable of reproducing the numerical results reported in the paper.

---

### Meta-Review · Area_Chair_D3d2 · 2024-12-21

**Metareview:**

This paper proposed a learning based large neighborhood search (LNS) method to solve large-scale Mixed Integer Program (MIP) problems. Key contributions include: 1) a binarized tightening technique to handle general integer variables, 2) an attention-based tripartite graph to capture global correlations among variables and constraints, and 3) an extra branching network trained supervisedly to override potentially incorrect LNS variable fixing actions. Reviewers acknowledged the novelty of the proposed method, especially the binarized tightening technique, which clearly distinguish it from existing works. They also appreciated the good empirical performance. One limitation is the requirement of high-quality solutions as training labels. Nevertheless, the lable collection cost is restricted to a small amount of time which is reasonable. Another limitation is that some key points in the methodology are not presented clearly. Overall, this paper is interesting and makes non-trivial contribution to learning based MIP solving. I urge the author to incorporate all comments in the revision, and open source their code for reproducibility.

**Additional Comments On Reviewer Discussion:**

Authors provided detailed point-to-point responses to reviewers' comments, with additional results and comparison to an contemporary work. Though one reviewer did not response to authors' rebuttal, I think the authors did a good job in addressing their concerns.

---

### Decision · Program_Chairs · 2025-01-22

Accept (Poster)